# Holistic face recognition is an emergent phenomenon of spatial processing in face-selective regions

Sonia Poltoratski [1✉], Kendrick Kay [2], Dawn Finzi [1] & Kalanit Grill-Spector [1,3]

Spatial processing by receptive fields is a core property of the visual system. However, it is unknown how spatial processing in high-level regions contributes to recognition behavior. As face inversion is thought to disrupt typical holistic processing of information in faces, we mapped population receptive fields (pRFs) with upright and inverted faces in the human visual system. Here we show that in face-selective regions, but not primary visual cortex, pRFs and overall visual field coverage are smaller and shifted downward in response to face inversion. From these measurements, we successfully predict the relative behavioral detriment of face inversion at different positions in the visual field. This correspondence between neural measurements and behavior demonstrates how spatial processing in face-selective regions may enable holistic perception. These results not only show that spatial processing in high-level visual regions is dynamically used towards recognition, but also suggest a powerful approach for bridging neural computations by receptive fields to behavior.

---

[1] Department of Psychology, Stanford University, Stanford, CA, USA. [2] Center for Magnetic Resonance Research (CMRR), Department of Radiology, University of Minnesota, Minneapolis, MN, USA. [3] Wu Tsai Neurosciences Institute, Stanford University, Stanford, CA, USA. ✉email: sonia09@stanford.edu

Visual recognition is critical to human behavior, informing not only what we see, but also how we navigate, socialize, and learn. In humans and non-human primates, visual recognition is carried out by a series of hierarchical, interconnected cortical regions spanning from the occipital pole to the ventral temporal cortex (VTC[1–3]). At the early cortical stages of this ventral visual stream, neurons respond to spatially local regions in the visual field and are typically tuned to simple stimulus properties like orientation or spatial frequency[4]. Later regions in the hierarchy show selectivity for complex stimuli like faces or places, and their responses correspond to perceptual experience[5,6]. These high-level responses are thought to be abstracted from and largely invariant to low-level image properties like stimulus size or position in the visual field[7,8]. However, modern research suggests that neural responses in VTC are sensitive to both stimulus size and position[9–14], providing a key empirical challenge to classical theories. Yet, it remains unknown if and how spatial processing in high-level visual regions is used toward recognition behavior.

The basic computational unit of sensory systems is the receptive field (RF), the region in space to which a neuron responds[4]. Throughout the visual system, neurons with similar RFs are clustered in cortex, allowing researchers to use fMRI to measure the population receptive field (pRF), or the portion of the visual field that is processed by the population of neurons in a voxel[15]. Recently, our group and others have used pRF modeling to quantify spatial processing in face-selective regions, finding that pRFs in ventral face-selective regions are progressively larger than in earlier visual areas and are densely centered around the center of gaze (the fovea), resulting in foveally biased coverage of the visual field[16–18]. These response properties may enable neural populations in face-selective regions to integrate information across facial features for effective face recognition. pRFs and their collective visual field coverage in face-selective regions also become larger and more foveal across childhood development, alongside with changes in fixation patterns and improved recognition of faces[17]. These findings lead to the intriguing hypothesis that spatial processing in high-level visual regions may enable and constrain recognition behavior. Here, spatial processing refers to how underlying neural populations sample the visual field, and over what extent they process visual information.

Human face recognition provides an ideal model system for deriving links between neural computations and behavior. More than two decades of research have characterized the core human visual face network in the brain[19] (Fig. 1a), defined as four anatomically distinct functional regions: one in the inferior occipital gyrus (IOG-faces/OFA), one on the posterior fusiform gyrus (pFus-faces/FFA-1), one straddling the mid-fusiform sulcus (mFus-faces/FFA-2), and one on the posterior end of the superior temporal sulcus (pSTS-faces). This network provides a compelling testbed for hypotheses relating behavior to neural computation, as it is reliably localized in each human participant, causally involved in face recognition behavior[20,21], and stereotypically organized in relation to known anatomical landmarks and cytoarchitectonic regions[22]. Importantly, the face recognition behavior supported by these regions has likewise been richly studied across decades of research[23,24]. Of particular interest here is the behavioral face inversion effect, or FIE[23]. First described over 50 years ago, the FIE is a robust and pronounced behavioral deficit in recognizing faces when they are presented upside down. Critically, the FIE has been attributed to a failure in spatial integration of information across multiple features of an inverted face[24–26], thus providing a behavior that can be directly linked to spatial computations in high-level visual regions.

In the current work, we hypothesize that spatial processing in face-selective regions provides the basis for integration of information toward face recognition behavior. Since face recognition is thought to be instantiated by the aggregate population of neurons spanning a region[27–29], this leads to the prediction that face inversion, which is thought to hinder typical spatial processing of faces[23,26,30,31], may alter pRF estimates and visual field coverage in face-selective regions, but not in early visual regions where spatial processing is not sensitive to face content.

To test our hypotheses, we map pRFs across the core face network with both upright and inverted faces (Fig. 1b). We predict that if large and foveal pRFs in face-selective regions are adaptive for typical recognition behavior, both individual pRFs and the combined visual field coverage afforded by the neural population of an entire cortical region may be shifted in position and/or in size with face inversion. If spatial processing in face-selective regions is instead a concomitant neural property that does not actively play a role in face recognition, we would expect that pRF estimates and visual field coverage when mapping with upright or inverted faces in both face-selective regions and early visual cortex would be equivalent.

Finally, we ask if spatial processing in face-selective regions plays a role in holistic face recognition, using the FIE as a test. Specifically, we reason that if visual field coverage in face-selective regions is reduced or shifted in response to face inversion, then neural processing of inverted faces at the center of the visual field would be suboptimal, and behavioral recognition of inverted faces would be impaired. Further, we conjecture that the behavioral FIE may be mitigated if faces are shown in the optimal location for inverted faces as determined from the visual field coverage they elicit in face-selective regions. As we find that visual field coverage in face-selective regions is smaller and shifted in response to face inversion, we use our participants' neural measurements to predict locations in the visual field where the magnitude of the FIE would be diminished. We then test these predictions in a separate behavioral face recognition experiment conducted outside of the scanner.

## Results

To measure spatial processing and estimate population receptive fields in response to upright and inverted faces, 12 participants took part in an fMRI experiment. Faces were presented in randomized order at 25 locations spanning the central ~9.2° of the visual field while participants maintained fixation on a central letter stimulus (Fig. 1a; see Methods). During pRF mapping, participants performed a challenging rapid visual stimulus presentation (RSVP) letter task at fixation. While participants were aware that faces would appear on the display, they were instructed that faces were not task relevant during the fMRI session. We saw no differences in behavioral performance on the letter task between trials in which upright (90.2 ± 1.8% correct) or inverted (89.4 ± 2.5%) faces were presented (post-hoc $t(11) = 0.531$, $p = 0.606$). Therefore, we infer that any observed differences in pRFs measured in response to upright and inverted faces are stimulus driven by the face inversion, rather than by task performance or attention. Additionally, eyetracking performed concurrently with fMRI confirmed that participants were able to maintain stable central fixation in each of the experimental conditions (Supplementary Fig. 1).

We first measured responses in each voxel to upright and inverted faces in 25 locations across the visual field. Notably, in face-selective cortex, responses to faces across the visual field positions were substantially different for upright and inverted faces. For example, responses of the mFus-faces voxel shown in Fig. 1e, left panel are lower in amplitude and shifted in the visual field when the mapping faces were inverted. Both the amplitude reduction and lower responsivity in the upper visual field are

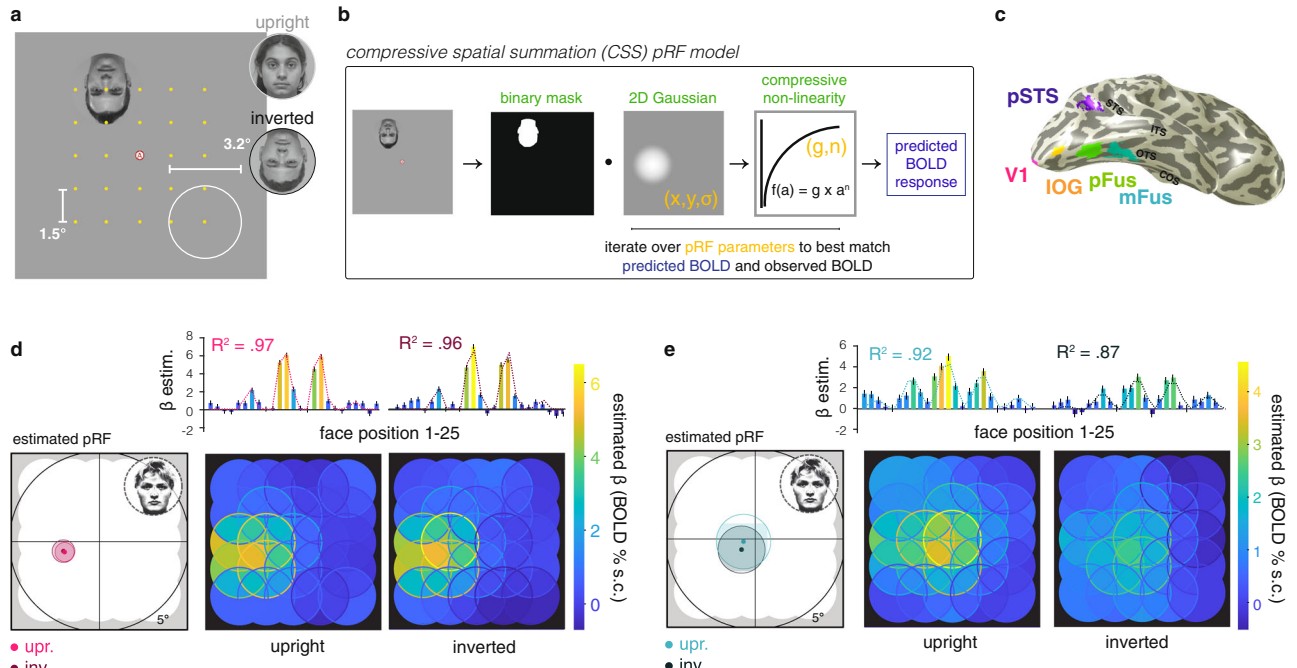

**Fig. 1 Experimental design and pRF model fitting procedure. a** Schematic illustrating the 5 × 5 grid of positions (yellow dots, 1.5° center-to-center) at which faces (3.2° diameter) were presented, either upright or inverted as shown. **b** Model fitting procedure of the compressive spatial summation (CSS) pRF model, which represents the position of each face in the visual field as a binary mask derived from the face silhouette. Estimated parameters (yellow text): center position X° and Y°, spread σ°, exponent (n), and gain (g). While a single binary mask is shown here, an averaged silhouette mask of the faces that each participant saw at each position was used for the model fitting. **c** V1 and face-selective regions of interest on the cortical surface of a representative participant. Acronyms: V1: primary visual cortex; IOG: inferior occipital gyrus face-selective region, also referred to as the occipital face area (OFA); pFus: posterior fusiform face-selective region, also referred to as the first fusiform face area (FFA-1); mFus: mid-fusiform face-selective region, also referred to as the second fusiform face area (FFA-2); pSTS: posterior superior temporal sulcus face-selective region. **d**, **e** Example V1 and mFus-faces single-voxel data and model fits. Top (bar plots): BOLD responses (β estimates in % signal change) for each of the 25 mapping positions for this voxel, plotted first from left to right, and then top to bottom. Error bars: ±SEM of the β estimates. Dotted lines: pRF model fit, which was estimated separately for upright and inverted faces. Left: estimated pRF in response to upright (light colors) and inverted (dark colors) faces. Circles contours indicate estimated pRF size, radius = σ° / √n. Right (overlapping-disk) panels: Responses at each position at which the mapping stimuli appeared, colored by response amplitude. Overlapping stimuli responses are averaged within each disc, while the absolute responses at each position are indicated by the disc outline. Upr.: mapping with upright faces, inv.: mapping with inverted faces. All individuals depicted provided informed consent for publication of their images.

visible in many individual voxels in face-selective regions (additional examples available on github.com/VPNL/invPRF). Consequently, these differences are also evident in the mean responses of face-selective regions at the 25 mapping locations (Supplementary Fig. 2). Importantly, we do not see substantial responses differences between upright and inverted-face mapping in V1 voxels (Fig. 1d, left panel), consistent with the idea that spatial processing in early visual cortex is not sensitive to face inversion. Given that the size, location, and individual faces used for upright and inverted faces mapping were identical, these differences in responses in face-selective regions suggest that face inversion yields specific differences in spatial processing.

To quantify these changes in spatial processing, we estimated the population receptive field (pRF) of each voxel separately for upright and inverted-face mapping conditions. We implemented the compressive spatial summation (CSS) pRF model[16,32] (Fig. 1b), and estimated for each voxel the model parameters that best explain its BOLD response amplitudes at the 25 mapping locations. The CSS pRF model explicitly considers the location and extent of the mapping stimuli, and yields estimates of five parameters that describe the gain (g, or amplitude of response), position (X,Y), and size ($\sigma/\sqrt{n}$[32]) of the pRF of each voxel, as well as a measure of the goodness-of-fit of the model ($R^2$).

pRFs were estimated independently for upright and inverted faces in each voxel in the primary visual cortex (V1) and four face-selective cortical regions (Fig. 1c). The CSS pRF model

accurately quantifies these voxels' responses, as it explains the majority of their respective variances (Fig. 1d, e-$R^2$). Resulting fits for the example voxels in Fig. 1 yield a precisely quantified summary of region in space that drives BOLD responses in the voxel. In V1, the independent model fits yield similar pRF estimates across upright and inverted faces (Fig. 1d). However, in the example mFus-faces voxel, the estimated pRF is smaller and centered lower in the visual field with face inversion (Fig. 1e).

**Mapping with inverted faces modulates pRF properties in face-selective regions but not V1.** To summarize pRF results observed for individual voxel fits, we quantified and compared average pRF properties for upright and inverted faces in face-selective regions. We found no significant differences in the horizontal position of pRF centers across upright and inverted mapping conditions in face-selective regions (Fig. 2a, Supplementary Table 1). However, face inversion yielded pRF centers that were consistently shifted downwards (Fig. 2b). We quantified changes in estimated pRF parameters using a three-way repeated-measures analysis of variance (ANOVA) with factors of ROI (IOG-/pFus-/mFus-/pSTS-faces), hemisphere (right/left), and mapping condition (upright/inverted). This revealed a significant effect of inversion on pRF center Y position ($F(1,10) = 8.82$, $p = 0.014$; no other significant main effects or interactions, Supplementary Table 1). Post-hoc $t$-tests (paired sample, two-sided in all reported comparisons) revealed that downward shifts in the average pRF

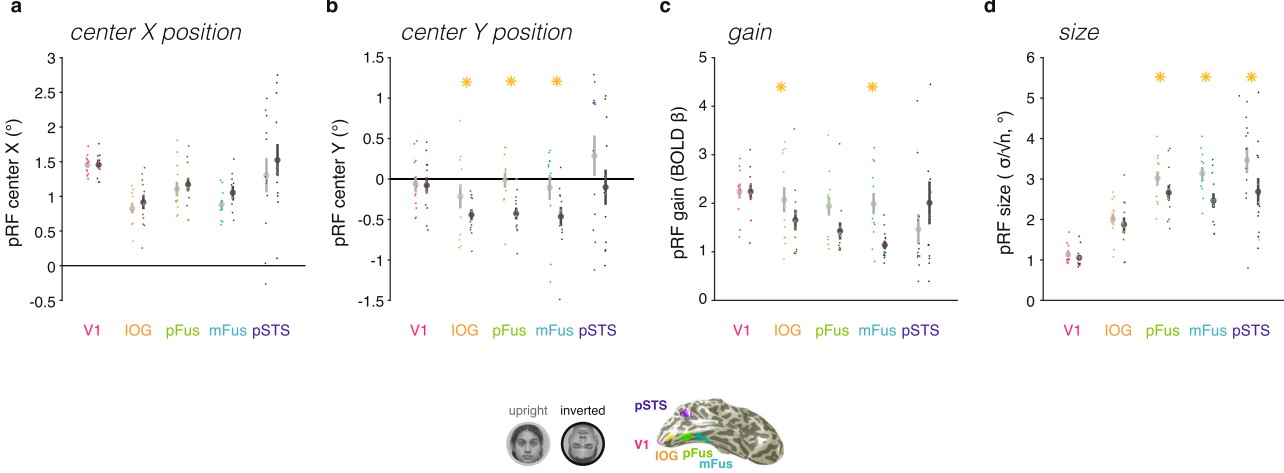

**Fig. 2 Face inversion impacts pRF estimates in face-selective cortical areas but not primary visual cortex.** Mean pRF parameters when mapped with upright faces (light gray) and inverted faces (dark gray) in bilateral regions of interest (ROIs). Gray markers: mean across 12 participants. Error bars: ±SEM across participants. Dots: individual participant means. Asterisks: statistically significant differences between upright and inverted-face mapping conditions (two-tailed paired post-hoc $t$-test across participants, $p < .05$ following analysis of variance (ANOVA); full statistics in Supplementary Table 1). **a** pRF center X° position; right hemisphere data are mirror flipped to left. **b** pRF center Y° position. **c** pRF gain. **d** pRF size ($\sigma° / \sqrt{n}$).

position were significant in all ventral face regions (IOG-, pFus-, mFus-faces, Fig. 2b). Additionally, pRF gain was significantly lower for inverted than upright faces in some of the face ROIs (Fig. 2c; significant ROI by inversion interaction $F(3,30) = 4.66$, $p = 0.0086$, three-way repeated-measures-ANOVA; no other significant effects, Supplementary Table 1). Post-hoc $t$-tests showed that pRF gains (which estimate the magnitude of responses) were lower in response to inverted than to upright faces in IOG- and mFus-faces (Fig. 2c). This finding is consistent with prior reports of lower mean responses for inverted than upright faces in ventral face-selective regions[30,33,34].

Interestingly, there were also significant differences in estimated pRF size across upright and inverted mapping conditions, which varied by face-selective region (ROI by inversion interaction $F(3,30) = 7.53$, $p = 6.7 \times 10^{-4}$, three-way repeated-measures-ANOVA, Supplementary Table 1). Specifically, pRF size was significantly smaller in fusiform face-selective regions (pFus- and mFus-faces) and pSTS-faces in response to face inversion (post-hoc $ts(11) > 2.98$, $ps < 0.013$; Fig. 2d), but not in IOG-faces.

Overall, face inversion had the most pronounced effect on pRF properties in mFus-faces, the end stage of the ventral face-processing hierarchy[18,19]. In mFus-faces, face inversion yielded a significant downward shift of pRFs, smaller pRF size, and smaller gain. As we did not find any significant difference across hemispheres on the effect of face inversion on pRF responses (Supplementary Table 1), we combine data across hemispheres unless noted. In contrast to the profound effect face inversion had on pRF properties in face-selective regions, we saw no significant differences between mapping conditions in any estimated pRF property in V1 (Fig. 2-V1). This suggests that spatial processing is altered in response to face inversion in higher-level face-selective regions, but not in early visual cortex.

**Signal-strength differences are insufficient to explain the observed effects.** As BOLD amplitudes were generally reduced in face-selective regions in response to inverted faces, we observed lower estimated pRF gain (Fig. 2c). Additionally, model goodness-of-fit in face-selective regions was consistently lower in the inverted than in the upright mapping condition (Supplementary Fig. 3). We asked if these reductions in signal strength might explain the observed differences in pRF position and size.

To test this possibility, we simulated the effect of changing the magnitude of responses and level of noise in mFus-faces on model estimates of pRF center Y position and size. Results of the simulations suggest that differences in signal-strength differences are insufficient to account for the scale of our observed effects in mFus-faces (Supplementary Fig. 4b, c).

**pRFs in face-selective regions are smaller across eccentricities in response to inverted faces.** The analyses of mean pRF size are consistent with the hypothesis that face inversion yields more spatially constrained integration of information via pRFs in face-selective regions. However, as pRF size linearly increases with eccentricity throughout the visual hierarchy[15,16,35], it is critical to consider the relationship between size and eccentricity when evaluating pRF size estimates across conditions. Thus, we compared the size vs. eccentricity relationship for upright and inverted faces in each ROI. We reasoned that if there are differences in spatial processing across mapping conditions, we should observe a higher intercept or a steeper slope relating size to eccentricity for pRFs mapped with upright than with inverted faces.

As evident in Fig. 3, we observed smaller pRF sizes in response to inverted than to upright faces across eccentricities in pFus- and mFus-faces. In contrast, we saw minimal differences in pRF sizes across eccentricities between upright and inverted-face mapping for either IOG-faces or V1, consistent with data in Fig. 2d. To evaluate these results across participants, we performed an equivalent line-fitting separately for each participant (Supplementary Fig. 5) and tested for significance. Results indicate the slope relating pRF size and eccentricity is not significantly affected by face inversion ($F(1) = 0.036$, $p = 0.85$, two-way repeated-measures ANOVA on slope with factors of ROI (IOG-/pFus-/mFus-/pSTS-faces) and condition (upright/inverted; Supplementary Table 2). However, pRFs were significantly smaller across eccentricities (significant main effect of condition on intercept ($F(1) = 9.08$, $p = 0.012$, two-way repeated-measures ANOVA, Supplementary Table 2). Post-hoc $t$-tests revealed that these effects were significant in both pFus- and mFus-faces (pFus: $t(11) = 4.19$, $p = 0.0015$; mFus: $t(11) = 2.37$, $p = 0.037$), but not in IOG-faces, pSTS-faces, or V1. We note that the CSS pRF model includes a compressive exponent (n) that is factored into the quantification of pRF size $\sigma/\sqrt{n}$. Additional analyses revealed

that the pRF size reduction with inversion is primarily driven by a decrease in the σ parameter (Supplementary Fig. 6), and is also observed for a linear pRF model without a compressive exponent (Supplementary Fig. 7; Supplementary Table 3).

**pRFs in face-selective regions are shifted downward in response to inverted faces.** To further explicate the effect of face inversion on pRF position in face-selective regions, we calculated the distributions of pRF positions in each participant and ROI when mapped with upright and inverted faces (Fig. 4a, Supplementary Fig. 8). In IOG-faces, the average distribution of pRF center Y positions mapped with upright (top panel) and inverted (bottom panel) faces were largely equivalent. In contrast, face inversion yielded a robust downward shift in pFus-, mFus-, and pSTS-faces (Fig. 4a, Supplementary Fig. 8b). In fusiform face-

selective regions, inversion shifted the overall distributions' centers from the horizontal meridian to the lower visual field. These shifts are visible when plotting the location of each voxel's upright vs. inverted pRF position as shown in Fig. 4b for mFus-faces. While we also see some variability in the estimated pRF center X position across upright and inverted mapping, these shifts have no consistent direction (Supplementary Fig. 8a), even when considering hemispheres independently (Supplementary Table 1, Fig. 2a). The downward shift in pRF center position in response to inverted faces altered the overall distribution of pRFs to be further from the fovea. Notably, increased eccentricities would have predicted larger pRFs mapped with inverted than with upright faces, which is the opposite of our empirical findings (Fig. 3).

**pRF changes result in differences in the coverage of the visual field in face-selective regions.** Given our findings that pRFs in face-selective regions are modulated by face inversion, we next asked how these changes may impact the overall coverage of the visual field afforded by neurons in these regions. This visual field coverage reflects the distributed spatial responsivity of the entire population of neurons spanning a cortical region, and thus may be more closely linked to behaviors that the population subserves. Thus, we constructed and quantified the density coverage map in each ROI for each participant and then averaged across participants. The density coverage map visualizes the mean proportion of each subject's pRFs that overlap with each point in retinotopic visual space (see Methods, Fig. 5). We calculated the area of the full-width half maximum (FWHM) of this mean density map as a summary metric of the field-of-view afforded by each ROI.

When pRFs are mapped with upright faces, ventral face-selective regions show a prominent foveal bias in coverage (Fig. 5, left-U). This foveal bias is markedly reduced in response to inverted faces (Fig. 5, left-I), and is coupled with a significant reduction in the coverage area, as well as a downwards shift of the coverage (Supplementary Table 3, Supplementary Fig. 9). These effects appear largely consistent across individual hemispheres (Supplementary Fig. 10). In face-selective areas, we observed significant main effects of condition (upright/inverted) on the coverage area and center-of-mass vertical (Y) coordinate (Supplementary Table 4; two-way repeated-measures ANOVA; $F$'s$(1,10) > 7.20$, $p$'s$ < 0.022$) but no significant interaction between condition and ROI on pRF coverage metrics (F's(3,30) $< 1.62$, $p$'s $> 0.20$). As in voxelwise analyses, mFus-faces shows the strongest differentiation between upright- and inverted-mapped

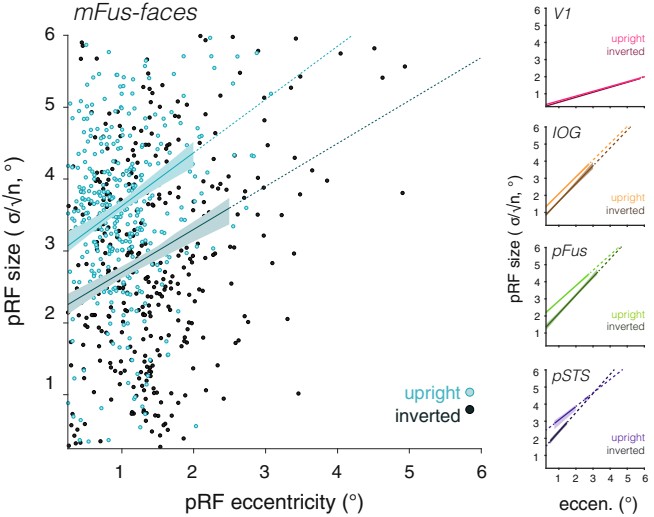

**Fig. 3 Face inversion yields smaller pRFs in face-selective regions across eccentricities.** Scatterplots showing the relationship between pRF size and pRF eccentricity for voxels in each ROI, mapped with upright faces (light colors) and inverted faces (dark colors). This relationship is summarized by the line of best-fit across all voxels pooled over participants with model goodness-of-fit $R^2 > .5$. Shaded region: a bootstrapped 68% confidence interval at 0.25° wide bins for which there were more than 20 voxels across participants. Dashed lines: bins for which there were fewer than 20 voxels across participants. Dots: a random sample of 300 mFus-faces voxels across participants. See also Supplementary Fig. 3.

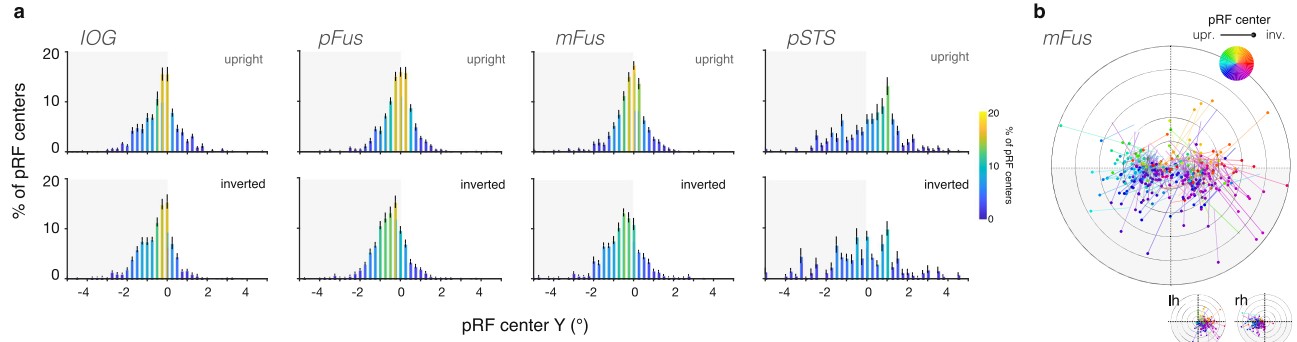

**Fig. 4 In face-selective regions, pRFs differ in their vertical position with face inversion. a** The average distribution of pRF center vertical position (Y) when mapped with upright (top row) and inverted (bottom row) faces. Distributions are computed for each participant, and then averaged across participants ($N = 12$). Error bars: ±SEM across participants in each 0.25° bin. **b** Position shifts for a random subsample of 300 mFus-faces voxels across participants for which the model variance explained is higher than 50% of the variance; voxel pRF center position estimates are consistently shifted downward.

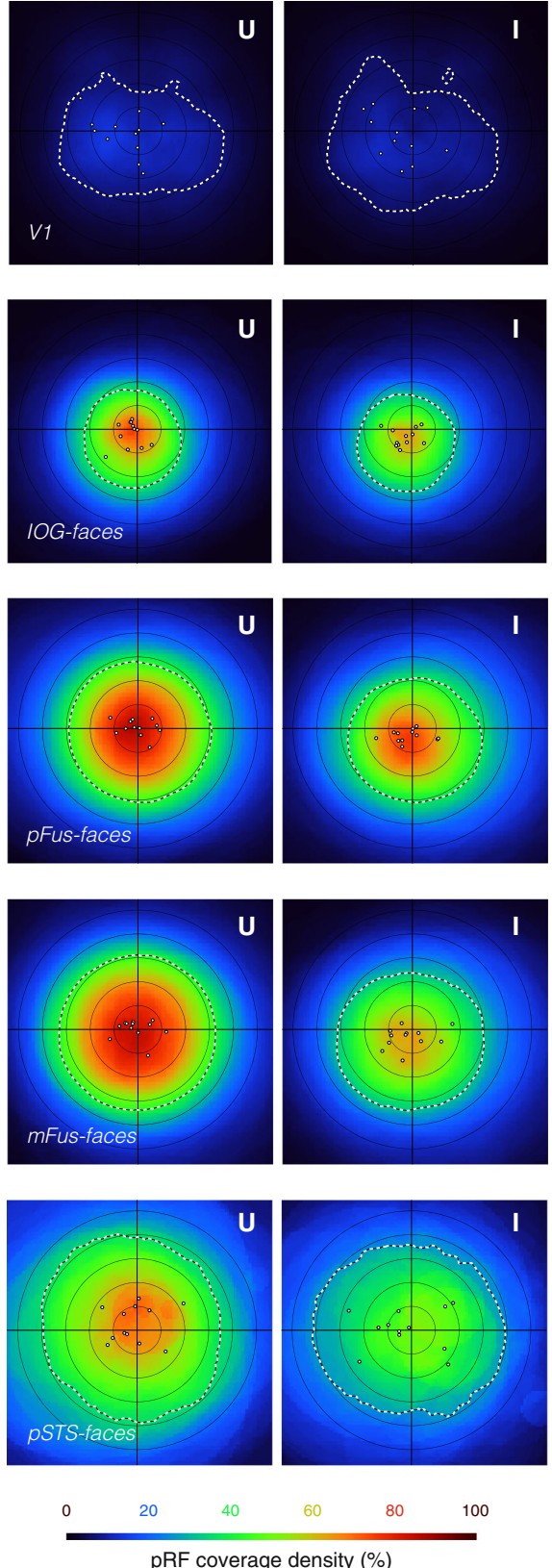

**Fig. 5 Face inversion leads to differences in visual field coverage of face-selective regions.** Average pRF density plots for each region across participants ($N = 12$) when mapped with upright (left) and inverted (right) faces. Density plots are bootstrapped for each participant using 1000 draws of 80% of the voxels and then averaged across participants (see Methods). The resulting plot color indicates the average percentage of voxels whose pRFs cover each point in retinotopic space, where coverage is defined as within $2\sigma/\sqrt{n}^{\circ}$ from the pRF center. Dashed white contours: full-width half maximum (FWHM) of the coverage density, the area of which is used as a summary of the overall visual field coverage of each region. Dots: coverage center-of-mass of each participant. U: mapping with upright faces; I: mapping with inverted faces.

center-of-mass X: post-hoc $t(11) = 0.57$, $p = 0.58$, Supplementary Fig. 7). A similar downward shift is evident in pFus-faces (significant change in center-of-mass Y: post-hoc $t(11) = 3.25$, $p = 0.0077$). In contrast, IOG-faces, pSTS-faces and V1 showed minimal differences in coverage between the mapping conditions.

Given prior reports of stimulus-dependence of visual field coverage in nearby word-selective regions[36], we sought to examine whether differences in coverage in face-selective regions were specific to face inversion. We evaluated this possibility by measuring the visual field coverage in our participants' face-selective regions to different stimuli: brightly colored cartoon images in a separate pRF mapping experiment (Toonotopy experiment[18], Supplementary Fig. 11a). Coverage in face-selective regions mapped with these cartoon stimuli was similar to the upright faces condition; pRF density was highest around the fovea, and consequently the coverage was centered on the fovea (Supplementary Fig. 11b-c). These data suggest that the downward shift of coverage observed in face-selective regions is specific to face inversion, and does not occur for any suboptimal mapping stimulus.

It is important to note that inherent to any pRF model implementation is the coding of the stimulus location in each timepoint. The standard pRF model[16,32], which we use here, codes the location of all visual stimuli as a simple binary mask image (Fig. 1b), and does not carry information about the stimulus features or content. Coding the stimulus differently—for example, preferentially coding high-contrast regions of the stimulus, or explicitly excluding the location of external features such as hair—will affect estimated pRF parameters (Supplementary Fig. 12). In particular, the pRF center position is closely linked to the center-of-mass of the coded stimuli (Supplementary Fig. 12a). While the standard pRF model does not capture the complex sensitivities to individual features[37] and face configuration[14,38,39] reported in face-selective regions, we can use alternative stimulus codings (Supplementary Fig. 12c) to evaluate the contribution of feature location to the reported differences between upright- and inverted-mapped pRFs. We note a correspondence between the magnitude of the observed vertical shift of pRFs in mFus- and pFus-faces with face inversion and the differential location of the internal features (Supplementary Fig. 12d). However, neither the location of the eyes nor of all internal features is sufficient to fully explain the observed effects of face inversion in face-selective areas (Supplementary Fig. 12d), and none of the tested alternative stimulus codings provided better fits to the data (Supplementary Fig. 12e). While future work should examine more complex representations of face features via weighting or configural templates[39,40], it is evident that a simple change to the manner in which the upright and inverted faces were coded is insufficient to account for the effects we observe in pRF estimates (Supplementary Fig. 2).

In all, we see profound differences in the way that face-selective regions sample visual space in response to face inversion, both at

visual field coverage (Fig. 5). Coverage for the latter spans a smaller portion of the visual field (significant change in full-width-half-max area: post-hoc $t(11) = 2.79$, $p = 0.017$) and is shifted downward relative to the former (significant change in center-of-mass Y: post-hoc $t(11) = 3.34$, $p = 0.0066$, but not

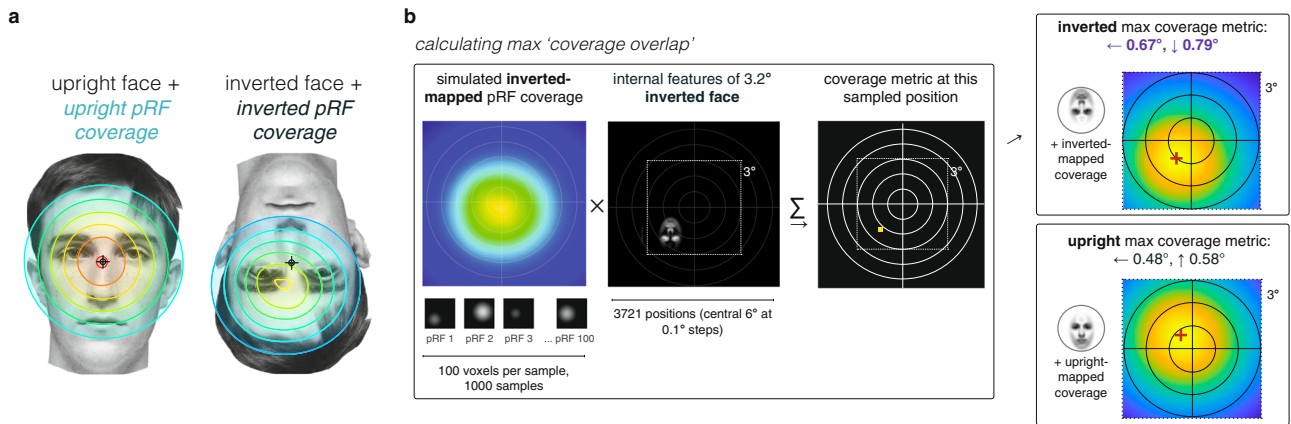

**Fig. 6 Using visual field coverage of face-selective regions to predict the location of improved performance for inverted faces. a** Schematized depiction of upright- and inverted-mapped mFus-faces pRF coverage when one views example faces centered on fixation. This predicts that viewing inverted faces yields suboptimal coverage of the location of the face features by pRFs in face-selective regions, which may contribute to the reported behavioral detriments in recognizing inverted faces (e.g., the face inversion effect, or FIE). **b** To evaluate the hypothesis schematized in **a**, we used a simulation procedure (see Methods) to determine the location in the visual field that would produce maximal pRF overlap with the internal features of 3.2° faces. To do so, we iteratively sampled a subset of mFus-faces voxels to generate bootstrapped coverage in response to inverted or upright faces; we then took the dot product of this bootstrapped coverage and a face centered at various retinotopic positions (central 3° × 3°, sampled at 0.1° steps). The retinotopic location that yielded the maximal overlap (dot product) between the inverted face and inverted-mapped pRF coverage of mFus-faces, as well as the upright face and upright-mapped coverage, is shown in the right panels. The individual depicted provided informed consent for publication of their images.

the level of individual voxels, and by the collection of voxels constituting a region. Importantly, pRF differences are observed even in response to passively viewed mapping faces, as participants' attention was diverted to an unrelated RSVP task on letters at the central fixation. This is particularly interesting as it suggests that the reported effects of inversion on spatial processing in face-selective regions is not a simple reflection of the participants' task or focus of attention, but instead reflect an underlying sensory property that may ultimately constrain behavior. To evaluate this idea, we next used our fMRI measurements to derive and test a specific prediction for participants' performance in recognizing upright and inverted faces outside of the scanner.

**Testing the relationship between pRF coverage and face recognition behavior.** During typical face viewing, pRF coverage in face-selective areas seems optimized to capture information from the face: large and foveal pRFs extend into the contralateral visual field, allowing for the processing of many features of a centrally viewed face by the pRFs of a face-selective region[16]. However, as schematized in Fig. 6a, our data suggest that this is not the case when viewing inverted faces. If participants fixate at the center of an inverted face in a standard recognition task, our data predict that the visual field coverage by pRFs of face-selective regions involved in recognition will suboptimally cover the inverted facial features useful for recognition (Fig. 6a). This leads to an intriguing hypothesis about the functional role of spatial processing in face-selective regions: if pRF coverage corresponds to windows of spatial integration toward face recognition, then increasing the overlap of this coverage with inverted facial features should yield better recognition performance of inverted faces, consequently reducing the magnitude of the behavioral face inversion effect (FIE) in that location in the visual field.

To evaluate this hypothesis, we used our pRF results to generate a specific prediction about behavioral face recognition performance. We consider the results of our pRF mapping during an irrelevant letter task at fixation to be a useful approximation of the neural 'hardware' on which task-based spatial integration might operate. To determine how visual field coverage might interact with demands of behavioral recognition, we first

implemented a simulation to predict the location at which the behavioral FIE may be attenuated (see Methods, Fig. 6b). While face recognition behavior is determined by a complex set of factors, including development across the lifespan[17,41,42], visual acuity[43], emotional valence[44–46], and familiarity[47–51], a large body of behavioral recognition and eyetracking research indicates that internal facial features are critical[23,24,52–55]. From this, we reasoned that face recognition would be best at the retinotopic location at which the internal facial features of the upright or inverted face maximally overlap with visual field coverage in face-selective regions mapped in the same orientation. Thus, we predicted that the FIE would be reduced if we showed faces at the location where inverted internal facial features maximally overlap with the region of densest visual coverage in response to inverted faces.

To determine this location, we simulated the overlap between the internal features of inverted and upright faces and the corresponding pRF coverage from mFus-faces, where the effects of face inversion were strongest. As in the fMRI study, we used 3.2° face images in the simulation, and determined that the optimal location to place these inverted faces was centered 0.67° leftward and 0.79° downward from fixation (Fig. 6b, top right). Notably, this location is different than the optimal location for upright faces, which is centered 0.48° leftward and 0.58° upward from fixation (Fig. 6b, bottom right). While in both cases, predicted performance is best in the left visual field[56,57], corresponding to known the lateralization of face-selective regions[58,59], only performance for inverted faces is predicted to be improved in the lower visual field (see also Supplementary Fig. 13).

We tested these predictions by measuring face recognition performance of each of our participants in a behavioral experiment conducted outside of the scanner. The behavioral study consisted of a challenging recognition memory task on upright and inverted faces (50% each, Fig. 7a). Fixation was held at a small central bullseye, and strictly monitored with an eyetracker (see Methods). On each trial, participants saw a sequence of three faces (all upright or all inverted), each shown for 400 ms (Fig. 7a). After an 800 ms interval, a target face was shown, and participants were asked to indicate whether the target

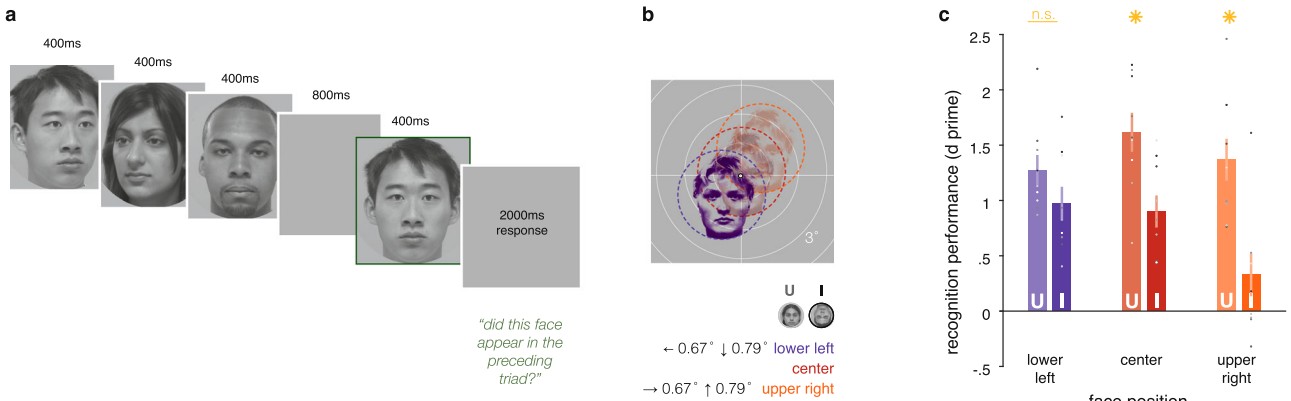

**Fig. 7 The behavioral FIE is reduced at the retinotopic position that maximally overlaps mFus-faces pRF coverage. a** Illustration of the trial structure and timing of the face recognition task outside the scanner. In the recognition task, participants viewed a triad of rapidly presented faces and indicated whether the subsequent target face had appeared in the triad. Within a trial, faces could be upright (as illustrated), or inverted, and appeared at the three locations described in **b**. Faces appeared at viewpoints ±15° around front-facing, and targets were always a different viewpoint than the triad exemplar. Subjects fixated on a central bullseye target (0.3° diameter), and their fixation was monitored with an eyetracker. Only trials on which participants maintained fixation were included in analyses. **b** Illustration of the three positions at which faces (3.2° diameter) appeared on the screen, as determined by mFus-faces pRF coverage from the fMRI experiment (Fig. 6b). **c** Recognition performance averaged across participants ($N = 9$) when they fixated and viewed upright faces and inverted faces in the three locations. Recognition performance for upright faces (light color bars), was predictably highest at the center (red), and decreased when the faces were presented in the lower-left (purple) or upper-right (orange) positions. However, recognition performance for inverted faces (dark color bars) was highest at the lower-left location consistent with the neural predictions. Across participants, the behavioral FIE (upright minus inverted performance) was significantly reduced in the lower-left position, while remaining strong at the center and upper-right positions. Error bars: ±SEM across participants. Dots: individual participant performance; Asterisks: significant paired two-sided post-hoc *t*-test differences between upright and inverted face conditions at $p < .05$. N.S. not significant; *t*-tests followed an ANOVA, see text. U: upright faces, I: inverted faces.

was present in the preceding triad (50% probability). Critically, to test our hypothesis, recognition performance on upright and inverted faces was evaluated at three retinotopic locations: the lower left, corresponding to maximal pRF overlap for inverted faces, central fixation, and an upper-right location that was equivalent in distance from fixation (Fig. 7b). This third upper-right condition serves as our primary comparison of interest, as it is matched in eccentricity to the critical lower-left condition while showing faces in the opposite side of visual space. Meanwhile, the central position provides a baseline estimate of the FIE at fixation, where it is it has been typically tested. We reasoned that if retinal acuity alone determines performance, recognition should worsen symmetrically with increasing distance from the fovea/center position for both upright and inverted faces. Notably, the lower-left/upper-right position shifts are relatively small: 3.2° faces are shifted 1.03° off-center in the lower-left and upper-right conditions. Individual trials on which participants broke fixation were excluded from the analysis, and the data of three participants who were unable to maintain consistent fixation were removed (see Methods).

We found striking differences in recognition performance at the three probe locations. Recognition for upright faces was best when faces were presented at the center of the screen and declined similarly when placed at the lower-left or upper-right positions (Fig. 7c-U). This is consistent with our predictions, as both the upper-right and lower-left positions were nearly equidistant from the simulated optimal position for upright faces (Fig. 6b; the lower left is 0.22° farther from the simulation's optimal position for upright faces). Recognition of inverted faces, however, followed a different pattern: performance was equal if not better at the lower-left location than at fixation, but starkly worse at the equidistant upper-right location (Fig. 7c-I). A two-way ANOVA with factors of position (lower-left/center/upper-right) and stimulus condition (upright/inverted) revealed significant main effects (inversion $F(1) = 35.76$, $p = 3.3 \times 10^{-4}$; position $F(2) = 10.37$, $p = 0.0013$) as well as a significant

interaction between position and inversion on face recognition performance ($F(2) = 4.57$, $p = 0.027$). While lateralization of face processing[58] would suggest a general improvement for both upright and inverted faces in the left hemifield[56], we instead see a specific advantage for recognizing inverted faces in the lower-left position. Indeed, the magnitude of the FIE, or the difference between recognition performance for inverted and upright faces, was not significantly different from zero at the lower-left location (post-hoc $t(8) = 1.68$, $p = 0.13$), while remaining significant at the other locations (center: post-hoc $t(8) = 3.43$, $p = 0.0090$, upper right: post-hoc $t(8) = 5.06$, $p = 0.0010$). Thus, the behavioral FIE is attenuated by placing faces in the location where inverted-mapped coverage of the visual field in ventral face-selective regions covers inverted facial features optimally. These results link the effect of face inversion on visual field coverage in face-selective regions to face recognition performance.

## Discussion

Our experiments demonstrate stimulus-driven changes of spatial processing in face-selective regions, quantified by the pRF model, which correspond directly to face recognition behavior. In the neuroimaging experiment, we demonstrated that face inversion altered spatial responsivity and coverage of the visual field in the face network, but not in primary visual cortex. While each of the face-selective regions showed some degree of sensitivity to inverted faces, effects were most pronounced in mFus-faces (FFA-2), where both individual pRFs and the coverage of the visual field were shifted downward and reduced in extent when mapped with inverted vs. upright faces. A similar pattern of results was found in pFus-faces (FFA-1). pSTS-faces, a region in the lateral stream[18,60,61] thought to process dynamic and social aspects of faces, exhibited similar effects of smaller pRF size, albeit noisier responses overall. This may be attributed to the substantially more peripheral responses in pSTS-faces[18,62], making our mapping with still images over a 9.2° × 9.2° visual field suboptimal for

this region. However, face inversion did not significantly alter visual field coverage in IOG-faces. This is consistent with the region's strong responses to individual face features[14,38], such as the eyes or mouth, on which inversion has a smaller perceptual effect. Importantly, we found differences in spatial processing in response to face inversion while participants performed a task that removed their directed attention from the faces. Following prior behavioral findings[63], this suggests that bottom-up properties of the stimulus itself, and not attention or task, drive the modulation of spatial responsivity by face inversion.

In the behavioral experiment, we demonstrated that the magnitude of the face inversion effect (FIE) varies across the visual field in a manner that is strikingly consistent with our measurements of differential visual field coverage in the fMRI study. Specifically, we showed that the magnitude of FIE can be mitigated by placing faces in an optimal location that corresponds to the inverted-mapped visual field coverage of mFus-faces. These data suggest that the FIE is a consequence of the differential population activity in face-selective regions evoked by face inversion, which corresponds to large-scale differences in the way these regions sample the visual field. More generally, our results highlight a precise relationship between spatial processing in high-level visual regions and the recognition behavior that they subserve.

**Holistic face processing as spatial integration across multiple features**. An extensive body of behavioral research has shown that faces are processed holistically—that is, face recognition behavior involves concurrent processing of the whole face, rather than its individual features[23,26,53,63–67]. However, despite decades of research into the behavioral signatures of holistic processing of faces, its underlying neural mechanisms remain opaque.

From our findings, we propose a hypothesis that holistic processing of faces can be understood as the spatial integration of information across face features, which emerges from how large and foveal receptive fields of ventral face-selective regions collectively process information across the visual field. Consequently, we suggest that behavioral failures of typical holistic processing, like the FIE[23], appear to be a result of maladaptive, stimulus-driven changes of pRF coverage in these regions. These findings provide neural support to cognitive theories that predict a reduced perceptual field for inverted faces[26]. Additionally, our findings suggest that deficits in spatial processing in face-selective regions may lead to impairments in face recognition as in congenital/developmental prosopagnosia[68–70], which can be tested in future research.

Importantly, our data not only elucidate the neural and computational mechanisms of face inversion, but also provide a computational framework that generates quantitative predictions of behavioral deficits of holistic processing. While our sample size did not allow us to probe individual differences in performance, this work provides an important foundation for future studies that can quantitively examine if individual differences in holistic processing or face recognition ability[55,70–73] are derivative of differences in pRF size, position, and coverage in high-level visual regions.

**Face inversion yields robust differences in spatial processing in face-selective VTC**. Our data demonstrate an interaction between estimates of pRF properties in face-selective regions and the stimulus used in mapping (upright vs. inverted faces). This appears specific to face inversion, as we did not find equivalent deviations in pRF estimates mapped with phase-scrambled faces[16] or diverse cartoons[18] (Supplementary Fig. 11).

What aspect of face inversion produces the observed changes in pRF estimates? While we considered the possibility that the differential location of the eyes or other internal face features contributes to our results, the location of these features alone cannot account for multiple differences in pRF estimates between upright and inverted faces, particularly in size and gain (Supplementary Fig. 12). Thus, while the current study highlights a critical component of spatial processing that contributes to visual recognition, any future unified model that accounts for responses to both upright and inverted faces will likely require incorporating additional factors including the level of responsivity to specific face features[14,38,37] and their spatial configuration[39,74] as well as other constraints such as visual acuity[43], crowding[75,76], and spatiotemporal capacity[77–79]. One recent proposal suggests that neurons in face-selective regions may also be modulated by contextual information about the typical configuration of faces above bodies[74]. This and related findings of learned selectivity for predictive spatial co-occurrences in ventral regions[40,80,81] may further elucidate the mechanisms underlying responses to upright and inverted faces across the visual field.

**High-level pRF processing as a bridge from brain to behavior**. While spatial sensitivity has been extensively demonstrated in VTC[9–13], the prevailing view theorizes that spatial processing in ventral temporal regions is independent from the recognition that these regions support. Our findings emphasize that high-level visual regions not only maintain spatially localized processing that can be computationally modeled by pRFs, but also that this processing dynamically supports visual recognition behavior. Doing so demonstrates the functional utility of spatial processing in the ventral 'what' stream.

More broadly, we view receptive fields (RFs) as a basic computational unit that is useful in many domains[82–86]. The results of this study highlight the fact that because neurons with similar RFs are spatially clustered in the millimeter range, pRF modeling can tap into complex, stimulus-dependent response properties of neural populations observable at the voxel level. We propose that quantitative measurement and modeling of pRFs in high-level regions across the brain opens compelling avenues for deriving precise links between brain and behavior—especially as activity in these regions often corresponds to perceptual experience more directly than in earlier stages of neural processing. In particular, our approach can not only quantitatively determine what factors may alter pRFs in other domains, but also leverage these measurements to generate new testable behavioral predictions.

The current work demonstrates that stimulus-driven changes in pRFs with face inversion, measured while participants perform an irrelevant task at central fixation, nonetheless impact behavioral recognition; these data suggest a bottom-up component of dynamic spatial processing in high-level VTC that may scaffold top-down and task-related activity. Future studies may incorporate additional task- and attention-based modulation of pRF properties into behavioral predictions. Our work[16] and others'[87,88] suggests that differences in pRF estimates between upright and inverted faces may be magnified during a face recognition task. However, future studies that vary behavioral task (e.g. recognizing aspects of identity[89,90], emotional expression[44,91], familiarity[92]) and stimuli (e.g., only internal face features, faces of different sizes) during pRF mapping are necessary to provide a more comprehensive understanding of the relationship between spatial processing in the ventral stream and face perception more broadly.

In sum, this work demonstrates a precise link between spatial processing in face-selective regions and recognition behavior. We

find that the neurally ill-defined concept of holistic face processing is an emergent property of spatial processing in high-level face-selective regions. These data suggest a powerful approach for bridging basic neural computations by receptive fields to behavior.

## Methods

**Participants**. Thirteen participants (six women) ages 20–31 participated in a single session of pRF mapping, including authors SP and DF. Nine participants identified as white/Caucasian, 1 as Asian, 1 as Black, and 2 as mixed-race (Hispanic/white and Asian/white). One participant's data was excluded for excessive motion (> 4 mm) during the scan. The studies conformed to all relevant ethical considerations for human participants research, as ascertained by the Internal Review Board of Stanford University. All participants provided informed, written consent and were compensated for their time. No statistical modeling was used to determine the size of our participant sample as we model each voxel individually in each participant. We used typical sample sizes for fMRI pRF studies of the human visual system, which use data from a range of 3–20 participants[15,16,35,36,87,93]. All scan participants completed the subsequent behavioral experiment.

**Scanning protocol**. Participants underwent a single fMRI session in the main experiment, which consisted of 8–10 runs of pRF mapping and lasted ~90 min. Scanning was done at the 3 Tesla GE research-dedicated magnet at the Stanford Center for Cognitive and Neurobiological Imaging (CNI). Functional MRI T2* weighted images were collected using a 32-channel headcoil, single-shot EPI, with a voxel resolution of 2.4 mm isotropic and a TR of 2 s (FOV = 192, TE = 30 ms, flip angle = 77°). Twenty-eight slices were prescribed parallel to the parieto-occipital sulcus to cover each participant's occipital and ventral temporal lobes.

We additionally acquired a high-resolution (1 mm isotropic voxels) T1-weighted full-brain anatomical image of each subject's brain either during the main experimental session or in a separate retinotopy/localizer session for each participant. We used this whole-brain anatomical image to align data across the main experiment, localizer, and retinotopy sessions in native participant space.

**pRF mapping**. Each population receptive field mapping run lasted 282 sec, and presented face images at 25 spatial positions to span the central 9.2° × 9.2° of the visual field over time (Fig. 1a). Face images were 3.2° in diameter, and were offset 1.5° center-to-center in a 5 × 5 grid arrangement. Faces were presented in 4 s trials (50 trials, 25 positions × upright/inverted) interspersed by ten 4 s blank periods per run, which were randomly ordered for each run and each participant. Each run began and ended with an additional 16 s blank period. Throughout each run, participants performed a challenging RSVP letter detection task at the central fixation point (0.4° diameter). Participants' fixation was monitored with an Eyelink 1000 eyetracker during the scan (Supplementary Fig. 1).

**Trial structure**. Each trial consisted of seven faces (300 ms on, 200 ms off) and one blank interval (500 ms) presented at a single spatial location. The same timing structure was used for the central letter presentation, including during the blank periods.

**RSVP task**. On each trial, participants were instructed to attend and respond to a stream of small, rapidly presented letters (A-S-D-F-G-H-J-K) and detect one-back repeats of the same letter. Half of all trials contained a repeat in the seven-letter stream, which occurred randomly in time within the trial. Letters were presented throughout the entire mapping run, and participants reported the task to be both difficult and attentionally engaging. Overall performance ranged from 77 to 98% correct (mean = 90%, SE = 2%), and we observed no difference between performance on trials that contained upright vs. inverted faces (post-hoc $t(11) = 0.531$, $p = 0.606$), confirming that participants' attention was not systematically drawn to either upright or inverted faces.

**Face stimuli**. We used front-view, gray-scaled face photographs previously used in Kay et al.[16] and other work in the lab (VPNL-101-faces). These depicted 95 different individuals of various genders and races in a neutral expression; demographics represent that of the Stanford University undergraduate student body. Faces were presented in a circular aperture and were matched on the overall height of the head, resulting in substantial consistency in the position of internal facial features and some variability in the position and presence of external features, including hair. The authors affirm that all individuals depicted provided informed consent for publication of the face images used in the Figures (Figs. 1, 2, 6, 7; Supplementary Figs. 6, 7, 9, 12).

**Experimental code**. The pRF mapping experiments and behavioral experiments were written as Matlab (2019a) code utilizing Psychtoolbox Version 3[94,95].

**Toonotopy and functional localizer experiments**. Each participant also completed an fMRI session of retinotopic mapping (four runs) using brightly colored, 8 Hz cartoon frames in a traveling wave aperture (Toonotopy[18]), as well as 3–10 runs of a functional localizer to identify category-selective regions (fLoc[96]).

**Data preprocessing**. fMRI data were preprocessed following a standard pipeline using FSL (https://fsl.fmrib.ox.ac.uk/fsl/fslwiki/Fslutils) and mrVista (https://github.com/vistalab) tools. Data from the experimental runs and the retinotopy and localizer analyses were co-registered to the participant's high-resolution T1. Functional data underwent motion correction (within- and between- scans), slice-time correction (pRF mapping only), and low-pass filtering (60 s). No spatial smoothing was done.

**ROI definition**. The localizer data were used to define face-selective regions in each participant by contrasting responses to images of child faces + adult faces versus images from eight other categories including bodies, limbs, objects, places, words, and numbers, with a threshold of $t \geq 2.7$, voxel level, uncorrected. Retinotopic regions V1-hV4 were defined from the Toonotopy data; boundaries between retinotopic areas were delineated by hand by identifying reversals in the phase of the polar angle map and anatomical landmarks, as detailed in ref. [97]. Face-selective region IOG was defined to exclude any voxels overlapping hV4.

**pRF model fitting**. We fit population receptive field (pRF) model estimates following the event-related paradigm introduced in Kay et al.[16]. This differs from many pRF implementations that fit the time course of responses to sweeping bars or rings/wedges (e.g., refs. [18,93]), and instead allows us to combine stimuli from different conditions (upright/inverted faces) in the same mapping run. mrVista functions (https://github.com/vistalab/vistasoft) were used to estimate GLM betas for each voxel in each of 50 conditions (25 locations × upright/inverted) using the SPM (http://www.fil.ion.ucl.ac.uk/spm) difference-of-gammas hemodynamic response function (HRF). Subsequent pRF modeling was done using these beta amplitudes.

Population receptive fields (pRFs) were fit independently to the data of each voxel in each mapping condition (upright/inverted) using the compressive spatial summation (CSS) model[32]. This models the pRF using a circular 2D Gaussian pRF followed by a compressive power-law nonlinearity (exponent $n < 1$). The nonlinearity decreases position sensitivity within the pRF and better fits neural responses in high-level visual areas[32] including face-selective regions[16]. We estimated pRFs independently in each voxel and for each mapping condition by fitting six model parameters: gain, center X, center Y, σ, and n. pRF size is defined as σ/√n, and visualized in Fig. 1d,e with a contour at radius = 2 × size from the pRF center. Model fitting was performed using nonlinear optimization (MATLAB Optimization Toolbox). Several bounds were imposed on the model fitting: the center X/Y position could not be outside the range of the mapping stimulus (11° × 11°); the σ could not be less than 0.1° or greater than 11°; gain was bound between 0 and 100; and the exponent was compressive between 0 and 1. These bounds were imposed to exclude values that typically corresponded to voxels not responsive to our stimuli (i.e., uniformly low responses across the visual field).

For quantification of pRF properties, we selected voxels with a goodness-of-fit $R^2 > 0.2$ in both the upright and inverted conditions; in visualizations that pool voxels across participants (Fig. 3), a threshold of $R^2 > 0.5$ was used for better visual clarity.

In the main experiment, face position was coded using a binary mask determined by the silhouette outline of the faces. These masks were drawn by hand for each face stimulus using Adobe Photoshop, and the model fit was based on an average of the specific faces that the individual participant saw at each of the 25 positions during their experimental session.

**Rescale + noise simulation (pRF experiment)**. To elucidate how differential signal strength or model goodness-of-fit may have contributed to the observed differences in inverted- and upright-mapped pRF size and position estimates, we ran an iterative simulation on data from right hemisphere mFus-faces, pooled across participants. As shown in Supplementary Fig. 4, simulated data for each voxel was generated by taking beta estimates from the upright condition and matching model goodness-of-fit (noise step) and beta range (rescaling step) to that of the inverted condition. The noise step added Gaussian noise to the betas until the simulated goodness-of-fit $R^2$ was within .01 of the $R^2$ in the inverted condition. The rescaling step involved divisively scaling the range of the upright betas to match that of the inverted betas. Both steps were done iteratively to vary the noise level while preserving the prescribed scale. To summarize the results of this simulation and compare across upright, inverted, and simulated-data estimates, we used bootstrapping (1000 draws) to estimate the median and 68% confidence interval of each model parameter of interest (position Y, size, and gain).

**Quantifying pRF size vs. eccentricity**. pRF size linearly increases with eccentricity throughout the visual hierarchy[15,16,35]. Thus, we compared the size vs. eccentricity relationship for upright and inverted faces in each ROI. We performed this analysis in two ways: (i) pooling well-fitted voxels ($R^2 > 0.5$) across participants in each ROI (Fig. 3), and (ii) separately estimating the relationship for each participant and ROI

(Supplementary Fig. 5). The former produces more robust line fits, while the latter provides an estimate of between-participant variability. Fitting was done by minimizing the L1 norm of the residuals, e.g., the sum of the absolute values of the residuals. This solution was chosen for its robustness to outliers in the underlying data.

**Visual field coverage.** Visual field coverage density plots for each visual region in Fig. 5 and Supplementary Fig. 8 were generated first for each participant, and then averaged across participants. Created using a custom Matlab bootstrapping procedure, these density plots represent the proportion of pRFs in a region that overlap with each point in the visual field. Overlap is determined using a binary circular pRF at the estimated center, and with a radius of $2 \times$ pRF size ($2\sigma/\sqrt{n}$[32]), which captures ~86% of the total volume of the CSS pRF. This metric does not account for the Gaussian profile of individual pRFs, but allows for greater interpretability when combining data across pRFs and participants.

For each participant and ROI, density plots were generated by taking 1000 bootstrap samples of 80% of voxels with replacement. Each voxel was represented with a circular binary mask as described above, and coverage was computed by calculating the mean density across voxels for each bootstrap draw. The average of these bootstrapped images is taken as the density coverage for each participant. Participant-wise metrics, like FWHM area and center-of-mass (Fig. 5, Supplementary Fig. 9), are computed from these images. Averages across participant-wise images are taken as overall coverage density summaries (Fig. 5, Supplementary Fig. 8); no rescaling or normalization is done, such that plotting colors retain meaningful quantitative information about pRF coverage across the visual field.

**Maximal-overlap location simulation for behavioral experiment.** To evaluate the hypothesis that inverted-face recognition would be improved at the retinotopic location that produced maximal overlap with inverted-mapped pRFs, we ran a simulation using preliminary data from the first six participants in the main experiment (Fig. 6b, Supplementary Fig. 13). We pooled voxels from bilateral mFus-faces across participants, and then took 1000 random samples of 100 voxels to generate a simulated pRF coverage. We then took the dot product of this coverage with an averaged absolute-contrast image of the internal features of the mapping faces at 3.2° diameter (such that the full-face image, not the internal features, was 3.2°), positioned at 0.1° intervals spanning the central 3° × 3° of the visual field. For each simulated coverage map, this yielded an overlap metric at each of these 0.1° intervals (Fig. 6b, right panel), and the maximal value was chosen. This yielded an average maximal overlap when the face was positioned at 0.67° to the left and 0.79° below the center; the leftward shift is a consequence of the fact that the right mFus-faces is larger than the left mFus-faces in most participants, so that there are more voxels that have pRFs covering the left than right visual field (see also ref. [58]). For comparison, the same simulation using the internal features of an upright face and the upright-mapped pRFs yielded a maximal-overlap position at 0.48° to the left and 0.58° above the center. See also Supplementary Fig. 13, in which the maximal-overlap location is computed for each of the 12 subjects in the full experiment.

**Behavioral experiment.** Following the simulation described above, we sought to probe recognition performance for upright and inverted faces at three positions: the lower-left location determined to yield maximal overlap between inverted-face features and inverted-mapped mFus-faces pRFs, a mirrored upper-right location that is equivalently far from the center, but predicted to have much lower pRF overlap, and the center of the screen, where FIE is typically measured. The experiment followed a standard behavioral face inversion paradigm[23], using a challenging recognition memory task in which participants reported whether or not a target face appeared in a preceding triad (Fig. 7a). The experiments were run in the eyetracking lab, and participants' fixation was monitored with an Eyelink 1000 as described below. All 12 of the scan participants took part in this experiment after their scan session, and eye position was carefully monitored to ensure fixation. Data from three participants were excluded from the final analysis because they failed to maintain fixation on more than 25% of all trials. Performance was measured for each condition (location × face orientation) as a sensitivity index (d').

**Experimental design.** Prior to starting the behavioral experiment, participants received practice on the task with performance feedback. Following the practice period, each participant completed 18 blocks of 20 trials. Position varied across blocks, such that each position (lower-left, center, upper-right) was probed on six blocks. Within each block, half of the trials used upright faces and half used inverted faces; trial ordering (i.e., inversion) was randomized. In total, each condition (face orientation × position) was probed in 60 trials, and the experiment took ~30 min to complete.

*Face stimuli.* Face identities were the same as in the scan experiment. In the behavioral experiment, we used front-view faces and also ±15° viewpoint angle images of the same individuals to increase the difficulty of the task. We note that these full-face stimuli, which were used to better approximate face recognition in natural vision, propagate small differences in the relative eccentricity of internal face features between the upper-right and lower-left condition. While these

eccentricity differences may contribute to behavioral performance, they are not a parsimonious account of the observed results, as it predicts differential performance for recognizing upright faces at the two off-center conditions.

**Trial structure.** Each trial (Fig. 7) consisted of three faces presented for 400 ms, followed by an interstimulus interval of 800 ms, and a 400 ms target face. Participants were asked to respond via keyboard whether the target face, which was marked with a thin green outline, had appeared in the previous triad. Participants were given 2400 ms from the appearance of the target face to respond. Targets appeared on half of trials, randomly as the first, second, or third face in the triad. On trials when the target was absent, it was replaced by a distractor face that was matched in general appearance (e.g., blonde woman with long hair) to the target, as maximally allowed for our set of 95 face identities. Trials were controlled so that the target image and its incidence in the triad were always of different viewpoints. That is, an identical image was never used in the triad and as the target. Each trial was preceded by a 2 s fixation intertrial interval (ITI) during which eye position was recorded to be used toward drift correction.

**Eyetracking.** Eyetracking during the behavioral experiment was performed using an Eyelink 1000 with a sampling rate of 1000 Hz, and analyzed using a custom Matlab pipeline. Prior to starting the experiment, participants received practice on the task with both performance feedback and real-time eyetracking feedback: we plotted their current eye position as a series of red dots while they performed the practice task at each of the probe positions. Participants were required to achieve good fixation (<0.5° deviation across trials in the second half of the practice) before they could proceed to the main task. Additionally, to improve the validity of our eyetracking metrics, we performed drift correction during each ITI; participants were aware of this and were instructed to keep steady fixation even between trials. This allowed us greater confidence in the absolute values of the tracked eye position on our displays, which was contained in the central ~5° of the visual field in total. Following the experiment, we manually removed from the analysis any trials on which participants broke fixation, defining this in individual trial traces as >1 fixation positions, and/or increased deviation in eye position around a position. We excluded three participants for whom this happened on >25% of all trials. For the remaining nine participants, an average of 9.51% (s.d. = 4.65%) of trials across conditions were removed as a result of breaking fixation.

**Reporting summary.** Further information on research design is available in the Nature Research Reporting Summary linked to this article.

## Data availability
Data from the neuroimaging study, behavioral eyetracking study, and simulations have been deposited in our lab's GitHub repository https://github.com/VPNL/invPRF, are publicly available, and can be cited and accessed using https://doi.org/10.5281/zenodo.4910755[98]. Data underlying every figure, table, and statistical result in the manuscript are provided as Matlab (https://www.mathworks.com/) .mat files. These .mat files can be read by multiple publicly available and free software packages, including Octave (https://www.gnu.org/software/octave/index) and Python (https://www.python.org) as HDF5 (https://www.hdfgroup.org) files. A README.md file (https://github.com/VPNL/invPRF/blob/main/README.md) describes the organization of the data and code in the repository, and how to generate each main and supplementary figure using provided code. Raw neuroimaging data are available under restricted access for the data privacy of our participants; access can be obtained by request of the authors. Source data are provided with this paper.

## Code availability
All code used in the analyses, simulations, and visualizations of the paper is available at https://github.com/VPNL/invPRF[98].

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

## Acknowledgements
This work was supported by NSF BCS #1756035 to K.G.S. and S.P.

## Author contributions
S.P., K.K. and K.G.S. designed the experiment. S.P. and D.F. collected and analyzed data. S.P. and K.K. wrote the analysis pipeline. K.G.S. oversaw the data analyses. S.P., K.K., D.F. and K.G.S. wrote the manuscript.

## Competing interests
The authors declare no competing interests.
