## [Peer Review File · Nature Communications]

REVIEWER COMMENTS

Reviewer #1 (Remarks to the Author):

Poltoratski et al characterized spatial sensitivity profiles of fMRI voxels (population receptive fields, or pRFs) located within a series of face-responsive regions of interest across conditions in which participants viewed (but ignored) an upright face and an inverted face. In face-responsive regions, especially mFus/FFA-2, the authors found that pRFs shift downward, decrease in amplitude, and shrink in size when a participant is viewing an inverted face compared to an upright one. These changes are absent in primary visual cortex, ruling out a low-level stimulus-driven explanation, and cannot be recapitulated by simple explanations like a difference in SNR between face orientations. Moreover, the authors tested whether these pRF changes may have behavioral consequences by testing the strength of a face inversion effect measured via behavior outside the scanner when face stimuli were presented at a location predicted to best mitigate the adverse processing impact the pRF changes in mFus may incur. Indeed, participants performed better on the task when inverted faces were presented in this predicted screen location compared to an equivalent position on the opposite side of the fixation point. Overall, the authors conclude that these changes in pRF properties reflect an important computational role for pRFs in supporting face perception, and that pRFs can achieve better spatial integration (due to larger pRF size) when a participant views upright rather than inverted faces.

I think this is a well-executed study (with really elegant figures!), and reveals some interesting variations in visual response properties across ROIs that are selective for higher-order stimuli. The conclusions, if supported, are likely of interest to the readership of Nature Communications. However, I have some reservations about the interpretation of the data, and concerns about the experimental design of the behavioral experiment which dampen my enthusiasm for the report overall. I think some of these issues may be addressable with additional analyses of the current dataset and/or adjustments to the text. I describe these issues, as well as raise some minor points, below.

Major points:

1. Throughout the manuscript, the authors attribute some degree of agency to population receptive fields (e.g., line 12: “neural computations by receptive fields”, lines 36-37: “spatial processing by pRFs in high-level visual regions may support recognition behavior”. At least in my view, a receptive field isn’t an active component of visual/perceptual processing – it’s a property of a neuron that depends on the stimulation and modeling protocol used to measure it (as these authors have shown here and in other reports, e.g., Kay et al, 2015). This is of course even more true for population RFs, which are an aggregate model describing properties of measured vascular signals that pool over hundreds of thousands of neurons. They’re surely related to neural response properties, but they’re fairly far removed from the actual computational units – the neurons – themselves (e.g., pRF properties will necessarily depend on experimenter choices like voxel size). That said, I’m extremely supportive of work to characterize RF properties of voxels/neurons across stimulus and/or task conditions and relate these observations to behavior – my principal issue here is with the framing that the pRFs are ‘doing’ something. The neurons are doing some computation, the voxels reflect that computation in aggregate, and the pRF model quantifies/characterizes that computation. But the pRF is just a description – not an agent. Perhaps I’m missing something, but if not, I think a few minor text alterations could help clarify this relationship.

2. The authors use a compressive spatial summation (CSS; Kay et al, 2013) pRF model throughout the manuscript. However, the stimulus setup doesn’t seem ideal for this sort of model. The CSS model diverges from the classical linear model (Dumoulin & Wandell, 2008) by applying a static nonlinearity after convolving the pRF profile with the stimulus mask, but before convolution with the HRF (unnecessary in this case because beta values are used instead). While I suppose it is in principle possible to infer a compressive spatial nonlinearity by just moving a stimulus around, this model seems to be more often applied to experiments using stimuli that are varied in their size, which directly manipulates spatial summation (to name a few, the original Kay et al, 2013 report did a version of this, and the more recent Kay et al, 2015 paper from this group used multiple sizes of face stimuli; I acknowledge that other recent publications from this group use the nonlinear CSS model with stimuli that do not vary in their size: e.g., Gomez et al, 2018, Nat Comms; however I’ll also point to the Supplemental Experimental Procedures of Kay et al, 2015, pg 4: “Since the reduced number of stimuli in these experiments provide insufficient data to

reliably estimate the exponent parameter (n), the exponent parameter was set to a fixed value (0.2).”). I’m concerned that there may be the potential for tradeoffs or interactions between the best-fit sigma and n parameters under these relatively impoverished stimulation conditions – is this something the authors can address, either via simulations, or potentially by comparing results using a standard linear Gaussian model? Specifically – across conditions, does n change? Or sigma? Or both? And what happens when using a simpler model? One possibility is that face inversion impacts only the compressive nonlinearity (decreases the exponent), which would increase the ‘size’ of the voxel (σ/\sqrt{n}), without changing its sigma (gaussian std dev). This means that the voxel is responding more-or-less equivalently strongly to stimuli at a greater number of locations. “spatial summation” or “spatial integration” in this case is a sort of strange term for what the model is describing (to me) – it instead sounds more like “spatial tolerance”. Perhaps I’m misguided or misunderstand, but I’m curious if the authors could address this.

3. There are a substantially different # of voxels in LH/RH for areas like mFus (mentioned in lines 487-489) – could the behavioral result in Fig. 7 be due to stimulus on left vs right, rather than aligned w/ ‘best’ location predicted from RFs? The ideal experiment to get at this would have been to present the face in each of the 4 quadrants, that way both hemifield and ‘optimal position’ could be controlled for together. However, there may be another way to get at this – Figure S7 shows the individual subject-derived optimal locations (the behavioral experiment used the average of these maps across participants, Fig. 6). Are subj whose individual optimal location for inverted face processing aligns closest with the mean better at the task/show a larger reduction in FIE than the subj whose individual optimal location are more poorly aligned? I understand that the statistical power to get at this effect in the present dataset is likely minimal, but even a qualitative result could help address this issue.

4. The authors conclude that larger pRFs during upright vs inverted face viewing reflect improved “spatial integration” (e.g., lines 9, 51, 144, many others). Maybe I’m extremely dense, but I’m not sure how this is reflected in their data. Yes, pRF estimates are larger in size when viewing upright faces (but see point 2). But doesn’t this just mean that a voxel in mFus respond to a face of a constant size with a stronger amplitude over a slightly larger extent of spatial locations? On any given stimulation trial, the face is the same size and just varies in its location. How does the resulting pRF estimate – which is based on these simple discrete measurements – tell me anything about spatial integration properties? Couldn’t these results be individually consistent with an RF of constant size that shifts to directly cover the face stimuli when they’re upright, but not inverted? (and aggregating trials across all such shifts would produce data identical to these) In such a scheme there’s no difference in spatial integration I don’t think. I’m likely missing something here, and the authors can likely add to the Results/Discussion to help make this point more clear.

Minor points:

1. There are some minor issues with statistics through the report. In particular, I don’t see any correction for multiple comparisons (nor a justification for this absence; Figs. 2, S3, 5/S4). Additionally, a few tests seem to show significant effects in some cases, but not others, but no ANOVA interaction is tested (Fig. 7), or the ANOVA interaction is tested and found to be non-significant, but not reported in the main text (Fig. 5/S4/Table S3)
2. Perhaps the authors could include a difference histogram as a 3rd row in Fig. 4? This may better illustrate the distribution of voxel-wise changes in parameters across inversion conditions. Additionally, I couldn’t figure out what the color scale on the histograms/colorbar was meant to indicate – the authors may want to clarify this in the caption or figure itself.
3. To what extent does using face silhouette as stimulus aperture in the modeling procedures matter? Did the authors attempt modeling with a circular aperture and performing some type of parameter comparison? (or, alternatively, generate simulated data using a face-shaped aperture, then fit w/ a circular aperture and see what’s lost, and vice versa). I ask because it seems plausible, if unlikely, that using a face-shaped stimulus aperture could exacerbate ‘shifts’ in measured pRF position that may not actually occur. Clearly this isn’t a uniform problem, as this result is specific to face-selective regions, but it’s still possible that for voxels like those in face-selective regions (big RFs, foveal), differences between the aperture using during modeling between conditions could exacerbate changes in parameter estimates.
4. What constraints were imposed on model parameters? (I see $n < 1$, and then a subselection of voxels w/ best-fit params within a certain range, but no information about the other constraints imposed)
5. Fig. 5 (and S4) compare something called “FWHM” of pRF coverage maps across conditions –

what is this measure the full-width at half-max of? It looks like some ROIs have non-circular/symmetric dashed white lines, which makes it hard to intuit how this quantity is calculated from the data.

6. In Fig. 6/the setup for the behavioral experiment, the authors used a contrast mask based on internal facial features to generate the predicted coverage plots under each stimulus condition. I wonder why the internal feature map was used here rather than the facial silhouette used throughout for pRF modeling? Why not use the same feature map in the pRF modeling itself?

7. For the behavioral experiment, a few questions about the eyetracking procedures: (a) how is a fixation break defined? (dva threshold?) (b) because the positions were tested blockwise and drift correction was applied online, is it possible subj were just 'drifting' to fixate on the center of each face, even during the ITIs?

Reviewer #2 (Remarks to the Author):

Review for NCOMMS-20-41124-T 'Holistic face recognition is an emergent phenomenon of spatial integration in face-selective regions'

Poltoratski et al. investigate a timely question with broad relevance: what functional role (if any) does spatial sampling play for face processing in the ventral temporal cortex (VTC)? Traditionally, the VTC 'what' pathway of visual processing has been characterised as location invariant, but recent years have shown that this view is incorrect. Neural populations in face preferring and other areas have characteristic spatial sampling profiles. However, we don't yet know a lot about the functional consequences of this.

Here, the authors present findings suggesting that the spatial sampling of neural populations in face preferring areas change with the type of stimulus presented. Crucially, these changes in spatial sampling appear to be at the heart of both, holistic face processing and its breakdown in the face inversion effect (FIE). Specifically, the authors fitted voxel-wise population receptive fields (pRFs) and estimated the visual field coverage of face preferring areas using either upright or inverted faces as mapping stimuli. They found that pRFs and field coverage esp. in fusiform face areas were broader and more foveally centred for upright mapping stimuli. For inverted faces, the fitted pRFs were smaller and shifted downwards, effectively covering less of the upper visual field. Based on this result, the authors predicted that the perceptual FIE should be smaller when stimuli are slightly shifted towards the field coverage for inverted faces (i.e. the lower left). They report results pointing in that direction.

If these results hold, they reveal an important functional role for a novel aspect of spatial sampling in the VTC, an aspect of the visual brain that is virtually ignored by textbook models and still poorly understood. At the same time, they would provide a convincing mechanistic account of holistic integration in face processing and its breakdown in the FIE, a major contribution to a longstanding debate in psychology (c.f. Rossion, 2009).

In my view, the findings by Poltoratski et al. are potentially exciting for a broad range of researchers of the brain and perception. However, I have major concerns regarding methods, analyses and interpretation, which I will list below. I do hope the authors will be able to address these concerns with adequate controls, for which I will provide suggestions.

MAJOR

1) Shifting pRFs versus shifting mismatches between modelled and effective stimulus- or: What's in a face?

Perceptual studies of the FIE often use stimuli restricted to the 'inner face', an oval aperture covering an area from chin to just above the eyebrows, in which the eyes appear at about 2/3 of the overall height (e.g. Van Belle et al., 2010). As acknowledged by the authors, there is plenty of evidence suggesting that responses in face preferring areas are mainly driven by these inner

features.

However, the mapping stimuli used here showed a frontal view of more or less the whole head, in which the eyes appear at about 50% of the height. That is, the mapping stimuli extended superiorly to include the full forehead and hair. In the discussion, the authors mention this in passing and point to a supplemental analysis (figure S8) suggesting that the apparent shift in pRFs may be driven by this (but assert that the change in pRF size wouldn't). If the effective stimulus was limited to the inner face, in the inverted condition modelled stimulus locations will be shifted downwards relative to the effective stimulus. This would necessarily lead to the appearance of a downward shift of pRFs.

This seems a crucial point to me, which deserves much more attention. Doesn't this change the interpretation fundamentally? There may be no shift of pRFs at all, just an artefact of an inadequate model assuming forehead and hair drive responses as well as the inner face. To illustrate the point: If the stimuli would have included the neck instead of hair and forehead, the apparent shift of pRFs may have been the opposite (upwards for inversion). But in that case the apparent shift would not reflect a change in spatial sampling of neural populations, but a change in the offset between the modelled and effective stimulus.

If the effects reported by the authors are truly contributing to the FIE as reported in the literature, they should be observable for the same kind of inner face stimuli (e.g. Van Belle et al., 2010). The ideal way forward would be a control experiment using inner faces as mapping stimuli. Given additional scanning is resource intensive, an alternative suggestion would be to enter the effective stimulus into the model by restricting the apertures for the analysis accordingly. If the downward shift of pRFs vanishes, this aspect of the results has to be interpreted much more cautiously. Crucially, such an analysis would also help to verify the assertion that differences in gain and pRF size cannot be explained by this confound. Do these latter effects persist when entering apertures for the effective stimulus into the model?

2) Small apparent increases in pRF size may reflect slightly better fixation compliance in the inverted condition

Observers may have had a harder time complying with the central fixation instruction when upright (rather than inverted) mapping stimuli appeared in the parafovea. This may then lead to the appearance of larger pRFs in that condition. The authors argue that this is unlikely, because they observed no significant differences in fixation task performance. But given the crucial nature of this finding and the potential confound, I think an explicit analysis of fixation compliance would be important.

The methods state that fixation was monitored during the scan with an eyelink 1000. Was this data recorded? If so, the authors could compare the median location and spread of measured gaze locations between conditions. Was gaze position more variable in the upright condition? If so, does the size of this effect match that of the observed increase in pRF sizes? If gaze data was not recorded, the easiest way forward may be a behavioural control experiment in the eyetracking lab, using the same stimuli and task.

3) Behavioural experiment: recognition performance may be driven by the eccentricity of inner features

This is essentially the same concern as for the mapping stimuli above. Recognition performance presumably hinges on the inner features. Given the whole-head nature of the stimuli, the eccentricity of these features is reduced in the inverted compared to the upright condition when stimuli are shown at the lower left location. The authors could rule out this confound by repeating the behavioural experiment with inner faces (again, as in Van Belle et al., 2010).

Another suggestion would be to test the FIE for different stimulus sizes. If the breakdown of holistic processing in the FIE is indeed a consequence of smaller pRFs, a strong and counter-intuitive prediction would be that recognition performance for inverted faces is enhanced when

they are presented at smaller sizes (at least relative to upright faces). This suggestion may be particularly relevant in case the apparent downward shift of pRFs does not persist when modelling effective stimulus locations (see 1) above).

4) Behavioural experiment: crucial interaction hypothesis not tested

The authors claim that 'the behavioral FIE (upright minus inverted performance) was significantly reduced in the lower-left position' (legend of figure 7) and that they 'showed that the magnitude of FIE can be mitigated by placing faces in an optimal location' (discussion). However, as far as I see, this interaction hypothesis was not tested. Instead this interpretation seems to rest on the fact that the FIE was significant at the two control positions, but only a non-significant trend was observed at the shifted location. This is a statistical fallacy. Absence of significance is no significant evidence of absence (esp. when $p = .13$ at a sample size of $n=8$) and a difference in significance provides no significant evidence for a difference (e.g. Gelman & Stern, 2006, *The American Statistician*). The authors should explicitly test their interaction hypothesis (reduced FIE at lower left compared to other locations). It may be wise to precede this with a power calculation for such an interaction test and increase sample size accordingly.

A further concern is regarding the prediction itself. Wouldn't the hypothesis that the lower left is a better match for the inverted field coverage specifically predict increased performance in the inverted condition? Instead, there seems to be a decrease in the upright condition.

MINOR:

- Noise simulation: betas of the upright condition are scaled to match the observed range of betas for the inverted condition – please clarify whether this was limited to positive values or not and briefly explain the rationale for that
- Noise simulation: the observed range for the inverted condition includes both, signal and noise. However, the final range of the simulated betas - after iterative addition of noise - will be larger than that, correct? If so, please clarify how large the final difference in scale is. A circular procedure with multiple iterations of scaling and adding noise may achieve both, similar scale and noise levels. However, given the results in figure S2c, I don't think this is essential.
- Relation to optimal fixation behaviour: Please briefly discuss how your results relate to previous findings showing optimal fixation positions within faces (Peterson & Eckstein, 2012 PNAS), as well as a possible match between spatial and feature sampling in face areas (e.g. Issa & DiCarlo, 2012 JNeuro; de Haas et al., 2016 JNeuro)
- Stimulus familiarity: Please briefly discuss the familiarity of participants with the stimulus set and the hypotheses (esp. regarding authors participating in the behavioral experiment)
- Threshold: How was the inclusion criterion $R^2 > 20\%$ chosen?
- Line 413: What does '60s' refer to in the context of low-pass filtering? Do you mean high-pass filtering signal with a period shorter than 60s? Sorry, in case I missed something obvious.
- SI line 11 superfluous 'that' ('match that the range')
- SI lines 99 and 100 – text seems to swap 'left' and 'right' panels
- Reporting of decimal points for stats (t-vals etc.) is inconsistent (sometimes two, sometimes three)
- Figure 5 : please provide an indication of scale (e.g. line showing extent of 1 dva)
- Line 459 'e.g.' presumably is meant to say 'i.e.'

Point-by-point response to the reviewers' comments. Our responses are **marked in blue**, as are changes in the manuscript document.

REVIEWER #1

Remarks to the Author:

Poltoratski et al characterized spatial sensitivity profiles of fMRI voxels (population receptive fields, or pRFs) located within a series of face-responsive regions of interest across conditions in which participants viewed (but ignored) an upright face and an inverted face. In face-responsive regions, especially mFus/FFA-2, the authors found that pRFs shift downward, decrease in amplitude, and shrink in size when a participant is viewing an inverted face compared to an upright one. These changes are absent in primary visual cortex, ruling out a low-level stimulus-driven explanation, and cannot be recapitulated by simple explanations like a difference in SNR between face orientations. Moreover, the authors tested whether these pRF changes may have behavioral consequences by testing the strength of a face inversion effect measured via behavior outside the scanner when face stimuli were presented at a location predicted to best mitigate the adverse processing impact the pRF changes in mFus may incur. Indeed, participants performed better on the task when inverted faces were presented in this predicted screen location compared to an equivalent position on the opposite side of the fixation point. Overall, the authors conclude that these changes in pRF properties reflect an important computational role for pRFs in supporting face perception, and that pRFs can achieve better spatial integration (due to larger pRF size) when a participant views upright rather than inverted faces.

I think this is a well-executed study (with really elegant figures!), and reveals some interesting variations in visual response properties across ROIs that are selective for higher-order stimuli. The conclusions, if supported, are likely of interest to the readership of Nature Communications. However, I have some reservations about the interpretation of the data, and concerns about the experimental design of the behavioral experiment which dampen my enthusiasm for the report overall. I think some of these issues may be addressable with additional analyses of the current dataset and/or adjustments to the text. I describe these issues, as well as raise some minor points, below.

We thank the reviewer for his/her thoughtful assessment of our work, and for the positive words about its impact and interest.

MAJOR:

1. Throughout the manuscript, the authors attribute some degree of agency to population receptive fields (e.g., line 12: “neural computations by receptive fields”, lines 36-37: “spatial processing by pRFs in high-level visual regions may support recognition behavior”. At least in my view, a receptive field isn’t an active component of visual/perceptual processing – it’s a property of a neuron that depends on the stimulation and modeling protocol used to measure it

(as these authors have shown here and in other reports, e.g., Kay et al, 2015). This is of course even more true for population RFs, which are an aggregate model describing properties of measured vascular signals that pool over hundreds of thousands of neurons. They're surely related to neural response properties, but they're fairly far removed from the actual computational units – the neurons – themselves (e.g., pRF properties will necessarily depend on experimenter choices like voxel size). That said, I'm extremely supportive of work to characterize RF properties of voxels/neurons across stimulus and/or task conditions and relate these observations to behavior – my principal issue here is with the framing that the pRFs are 'doing' something. The neurons are doing some computation, the voxels reflect that computation in aggregate, and the pRF model quantifies/characterizes that computation. But the pRF is just a description – not an agent. Perhaps I'm missing something, but if not, I think a few minor text alterations could help clarify this relationship.

We fully agree with the reviewer's assessment that pRFs have no 'agency,' so to speak, but rather reflect and quantify the aggregate of a computation performed by individual neurons.

Action: We have removed text in the paper that unintentionally suggested otherwise, and have more precisely described what we aim to characterize and quantify via the pRF model. Please see pg. 4, lines 36-39; pg. 6, lines 107-112; pg. 17, lines 347-350.

2. The authors use a compressive spatial summation (CSS; Kay et al, 2013) pRF model throughout the manuscript. However, the stimulus setup doesn't seem ideal for this sort of model. The CSS model diverges from the classical linear model (Dumoulin & Wandell, 2008) by applying a static nonlinearity after convolving the pRF profile with the stimulus mask, but before convolution with the HRF (unnecessary in this case because beta values are used instead). While I suppose it is in principle possible to infer a compressive spatial nonlinearity by just moving a stimulus around, this model seems to be more often applied to experiments using stimuli that are varied in their size, which directly manipulates spatial summation (to name a few, the original Kay et al, 2013 report did a version of this, and the more recent Kay et al, 2015 paper from this group used multiple sizes of face stimuli; I acknowledge that other recent publications from this group use the nonlinear CSS model with stimuli that do not vary in their size: e.g., Gomez et al, 2018, Nat Comms; however I'll also point to the Supplemental Experimental Procedures of Kay et al, 2015, pg 4: "Since the reduced number of stimuli in these experiments provide insufficient data to reliably estimate the exponent parameter (n), the exponent parameter was set to a fixed value (0.2) ."). I'm concerned that there may be the potential for tradeoffs or interactions between the best-fit sigma and n parameters under these relatively impoverished stimulation conditions – is this something the authors can address, either via simulations, or potentially by comparing results using a standard linear Gaussian model? Specifically – across conditions, does n change? Or sigma? Or both? And what happens when using a simpler model? One possibility is that face inversion impacts only the compressive nonlinearity (decreases the exponent), which would increase the 'size' of the voxel (σ/\sqrt{n}), without changing its sigma (gaussian std dev). This means that the voxel is responding more-or-less equivalently strongly to stimuli at a greater number of locations.

“spatial summation” or “spatial integration” in this case is a sort of strange term for what the model is describing (to me) – it instead sounds more like “spatial tolerance”. Perhaps I’m misguided or misunderstand, but I’m curious if the authors could address this.

We thank the reviewer for an insightful consideration of the theory behind this model, and for suggesting useful complementary analyses that we have now included as new Figures S6 and S7.

The reviewer is correct that it is possible, in theory, for there to be some differences in the compressive term of the CSS model across experimental conditions. As he/she points out, this is partly compensated by the use of the expression σ/\sqrt{n} when quantifying pRF size, which allows more of an ‘apples-to-apples’ comparison when describing pRFs with different exponents. We also agree that a size-varying mapping stimulus would be more optimal for robustly estimating the compressive term of the CSS model; because our study did not do this, we have presented the compressive exponent estimates only in aggregate as part of the pRF size estimate.

To investigate these issues more deeply, we have performed two sets of additional analyses. In the new Figure S6, we show that in our model, the size reduction in response to face inversion is largely driven by changes in the σ estimate. A 2-way repeated-measures ANOVA across bilateral face-selective areas shows a significant main effect of mapping condition on σ ($F(1)=7.59$, $p=0.019$) as well as a significant ROI x condition interaction on σ ($F(3)=3.32$, $p=0.032$). There were no significant effects of stimulus condition nor ROI on the estimated exponent (condition, $F(1)=2.69$, $p=0.13$; ROI, $F(3)=1.85$, $p=0.16$; interaction $F(3)=1.36$, $p=0.27$). We note that while there is a nominal difference in the estimated exponent, it is in the opposite direction than would be necessary to drive the observed decreased size: while the exponent estimate trends to be lower in some face-selective regions in response to inversion, this would, by our σ/\sqrt{n} definition of size, produce larger pRFs in response to inverted faces. Thus, it appears that the compressive term in face-selective regions is not changed with inversion, while the estimated sigma of the pRF Gaussian is reduced, yielding smaller apparent pRF size (σ/\sqrt{n}).

Complementary to the additional quantification of the CSS model results, we have also implemented a linear, non-compressive pRF model (exponent = 1) as in Dumoulin and Wandell (2008). The results echo the main findings of the paper (**Figure S7**, **Table S3**): a 2-way repeated-measures ANOVA across bilateral face-selective areas reveals a significant main effect of mapping condition (upright/inverted faces) on the σ estimate of pRF size ($F(1,10)=45.36$, $p=5.1 \times 10^{-5}$). Other parameters of this model (X, Y, gain) likewise follow the reported pattern of differences between upright and inverted mapping that we observed using the CSS model (**Table S3**). From this, we conclude that the model choice (linear/CSS) does not drive the results of the paper.

Action: The revised SI now includes new **Figures S6** and **S7**, and new **Table S3**. We refer to these results on page 10 (lines 187-191) of the manuscript.

3. There are a substantially different # of voxels in LH/RH for areas like mFus (mentioned in lines 487-489) – could the behavioral result in Fig. 7 be due to stimulus on left vs right, rather than aligned w/ ‘best’ location predicted from RFs? The ideal experiment to get at this would have been to present the face in each of the 4 quadrants, that way both hemifield and ‘optimal position’ could be controlled for together. However, there may be another way to get at this – Figure S7 shows the individual subject-derived optimal locations (the behavioral experiment used the average of these maps across participants, Fig. 6). Are subj whose individual optimal location for inverted face processing aligns closest with the mean better at the task/show a larger reduction in FIE than the subj whose individual optimal location are more poorly aligned? I understand that the statistical power to get at this effect in the present dataset is likely minimal, but even a qualitative result could help address this issue.

While it is true that face-selective areas are reliably right-lateralized (Yovel, Tambini, and Brandman, 2008) and that this is associated with better recognition performance in the left hemifield (Gilbert and Bakan, 1973), lateralization cannot account for the observed differences in performance for upright vs. inverted faces in our data. We do not observe a general improvement in face recognition when faces are placed in the left visual field; rather, we see a specific improvement for inverted face recognition at the probed lower-left visual field position while recognition of upright faces is comparable in both right and left off-center locations. This is predicted by the simulation results in **Figure 6** (right panel), which estimate optimal performance for upright faces would also occur in the left hemifield, but at a point slightly higher in the visual field than for inverted faces. This upright-optimal point is roughly equidistant to the probed upper-right and lower-left locations, which may account for the pattern of performance observed for upright faces.

Given our N = 9 in the behavioral experiment, we did not pursue individual differences comparisons in this study. Nonetheless, we have explored the analysis suggested by the reviewer: we calculated the distance between each subject’s ‘optimal’ pRF location for inverted face recognition and the lower-left probe location, and correlated this to the FIE reduction at the lower-left position relative to center. We did not observe any reliable correlation ($r^2 = 0.12$, $p = 0.36$) in this measure, but attribute this to the low power that our study yields for estimating individual differences.

Action: The revised manuscript clarifies that the behavioral finding does not simply reflect face representation lateralization (see pg. 17, lines 335-337). The revised manuscript also discusses several approaches that might explore individual differences in further, larger-N studies (see pg. 19, lines 387-391).

4. The authors conclude that larger pRFs during upright vs inverted face viewing reflect improved “spatial integration” (e.g., lines 9, 51, 144, many others). Maybe I’m extremely dense, but I’m not sure how this is reflected in their data. Yes, pRF estimates are larger in size when viewing upright faces (but see point 2). But doesn’t this just mean that a voxel in mFus respond to a face of a constant size with a stronger amplitude over a slightly larger extent of spatial locations? On any given stimulation trial, the face is the same size and just varies in its location.

How does the resulting pRF estimate – which is based on these simple discrete measurements – tell me anything about spatial integration properties? Couldn't these results be individually consistent with an RF of constant size that shifts to directly cover the face stimuli when they're upright, but not inverted? (and aggregating trials across all such shifts would produce data identical to these). In such a scheme there's no difference in spatial integration I don't think. I'm likely missing something here, and the authors can likely add to the Results/Discussion to help make this point more clear.

Here, the reviewer raises important points regarding the theoretical interpretation of pRFs and spatial integration. We agree that it is critical to carefully define these terms and describe our interpretation of the results in the paper. To clarify, we used the term 'increased spatial integration' to convey that when viewing upright faces, the region of space to which neurons respond is apparently larger, allowing the congregate computations carried out by all neurons in these regions a larger field of view over which to process information. In other words, 'spatial integration' is essentially recapitulating the notion of a population receptive field. The surprising aspects of our results is that (a) inverted faces lead to responses that reflect a smaller, shifted field of view and (b) this appears to have behavioral consequences for the recognition of faces. Indeed, the behavioral act of holistic face recognition has been hypothesized to require the spatial integration of information across spatially-distinct facial features, which is more robust if more neural activity in response to a broader spatial extent is enabled.

A natural question raised by the reviewer is what mechanisms yield these apparent changes in pRF size. We speculate that the underlying causes of this apparent decrease in pRF size with face inversion may reflect learned selectivity for predictive or diagnostic spatial co-occurrences (Kay and Yeatman, 2017) or frames of reference (Tadin et al., 2002; Almeida et al., 2020) in individual neurons or in populations – although determining these mechanisms will require substantial further work. The key insights delivered by our current manuscript is that these changes in pRF sizes do in fact exist and correspond to behavioral consequences, thereby providing a link between neural representation and behavior.

Action: To disambiguate the usage of the term 'spatial integration', we have revised the text to use the term 'spatial processing' to refer to congregate neural computations, and 'spatial integration of information across features' to describe the hypothesized mechanism of behavioral holistic processing. This is done throughout the manuscript and in the title.

MINOR:

1. There are some minor issues with statistics through the report. In particular, I don't see any correction for multiple comparisons (nor a justification for this absence; Figs. 2, S3, 5/S4). Additionally, a few tests seem to show significant effects in some cases, but not others, but no ANOVA interaction is tested (Fig. 7), or the ANOVA interaction is tested and found to be non-significant, but not reported in the main text (Fig. 5/S4/Table S3)

Action: We have corrected and clarified the issues related to statistical reporting. In particular, we have:

- (1) Clarified in each figure caption that t-tests are post-hoc, as indicated in the main text of the paper. Throughout the manuscript, we rely on ANOVA testing across ROIs to establish statistical significance; we only use the post-hoc t-tests to clarify which regions show significant effects. As such, we do not believe multiple corrections to be warranted. We thank the reviewer for raising this issue, likely because we accidentally omitted a critical ANOVA test from the manuscript, described in (2).
- (2) We now include the ANOVA for the behavioral experiment (**Figure 7**) including a test of the interaction term (see pg. 17, lines 332-337). This was an accidental omission, as we had tested for this significant interaction (condition x location, $F(2) = 4.57$, $p = 0.027$) prior to running further t-tests in the original analysis code (behavior-analysis/fig7_prfRec2_analysis_plusQA.m), but by mistake failed to report this in the paper.
- (3) Clarified in the main text that the interaction term for the coverage metrics (**Figure 5**) is not significant, but that the main effects are significant only in mFus- and pFus-faces.

Additionally, while preparing this revision we noticed a typographic error in **Table S3 (Table S4)** in the revised manuscript, as identical values were listed in both columns (coverage FWHM area, left, and center of mass Y position, right). We have now corrected this error in the revision by listing the correct values. The pattern of statistical results remains unchanged.

2. Perhaps the authors could include a difference histogram as a 3rd row in Fig. 4? This may better illustrate the distribution of voxel-wise changes in parameters across inversion conditions. Additionally, I couldn't figure out what the color scale on the histograms/colorbar was meant to indicate – the authors may want to clarify this in the caption or figure itself.

Action: We have added a new supplementary **Figure S8** that visualizes the averaged distribution of voxel-wise changes in parameters across subjects. We have also added a label to the colorscale of **Figure 4**, which echoes the Y axis magnitude (proportion of voxels) of the plots.

3. To what extent does using face silhouette as stimulus aperture in the modeling procedures matter? Did the authors attempt modeling with a circular aperture and performing some type of parameter comparison? (or, alternatively, generate simulated data using a face-shaped aperture, then fit w/ a circular aperture and see what's lost, and vice versa). I ask because it seems plausible, if unlikely, that using a face-shaped stimulus aperture could exacerbate 'shifts' in measured pRF position that may not actually occur. Clearly this isn't a uniform problem, as this result is specific to face-selective regions, but it's still possible that for voxels like those in face-selective regions (big RFs, foveal), differences between the aperture using during modeling between conditions could exacerbate changes in parameter estimates.

Here, the reviewer (also see Point 1 from Reviewer 2) raises the question of the impact of the stimulus aperture choice on the model fitting and interpretation. We agree that this is an important issue in the interpretation of this and any pRF modeling effort.

The reviewer is correct to note that in the 2015 paper from our group, we used circular disk apertures rather than silhouette outlines to represent the location of face mapping stimuli. In that work, faces were rotated along the midline (0° , $\pm 15^\circ$, $\pm 30^\circ$, $\pm 45^\circ$) in different images sampled for each experimental condition, such that the circular aperture was more uniformly stimulated across the experiment. In the current work, we only showed frontal faces that have a more limited spatial extent. Thus, we modified the coding of the stimuli to better represent the location of the faces in the visual field and improve the modeling accuracy.

To the reviewer's specific question about comparing estimates between the silhouette and disk stimulus coding, this is an interesting issue that can be viewed through the general framework of the **Figure S12a-b (Figure S8)** in the original paper. It is true that the absolute estimate of pRF center position is closely linked to the center of mass of the stimulus coding. In other words, the use of a disk aperture as opposed to a silhouette may indeed yield uniform, systematic differences in estimated pRF position (primarily in the vertical direction). However, the choice of stimulus coding cannot account for differences in size or gain parameters across the upright and inverted conditions.

We evaluated this possibility first by computing center-of-mass estimates on our full set of images and comparing these estimates to a circular disk. Because of the cropping procedure of the mapping face images, which were matched for overall face size within the aperture, we see relatively little difference between the average center-of-mass of the face silhouette masks and the circular disk in the vertical direction: the average upright face silhouette center-of-mass is 0.044° lower than the disk center, resulting in a 0.088° offset between upright and inverted conditions. We additionally expect that a model fit using the circular disk coding would yield overall smaller pRF estimates (since the stimulus coding itself is larger), but that this would happen equivalently for upright and inverted faces. We confirmed these intuitions in a new analysis (**Figure S12c-e**), which presents estimated parameters using a disk (D) stimulus coding and several others, and underscores the direct relationship between stimulus coding and parameter estimates. Nonetheless, pRF center Y estimates were lower in the visual field for the inverted faces than upright in pFus and mFus-faces for both the disk and the silhouette coding of the stimulus (new **Figure S12d-top**), indicating that both models predict a downward shift of pRFs for inverted vs. upright faces. Likewise, for both disk and silhouette coding schemes our model estimates smaller pRFs sizes in pFus- and mFus-faces for inverted vs. upright faces, (**Figure S12d-middle**).

More generally, the reviewer is tapping into the critical issue that any pRF model contains, via its stimulus coding scheme, an assumption of the way in which the mapping stimulus drives spatial responses — whether that is equivalent across the image content (e.g. binary masks), or involves more complex sensitivity to contrast or features. The primary thrust of the present modeling effort was to test the specific hypothesis that the location of all visual information, specified via binary mask, is sufficient to characterize spatial responses in face-selective areas. Our results show that this is not the case at specific stages of the face-

selective hierarchy, providing insight into the evolution of spatial coding along the ventral visual stream.

Action: The revised manuscript includes detailed discussion of our aperture choice in a new section of the results (see pg. 13, lines 243-258), as well as a more explicit description of the relationship between stimulus coding and parameter estimates via a new analysis of multiple stimulus coding schemes (see **Figure S12c-e**).

4. What constraints were imposed on model parameters? (I see $n < 1$, and then a subselection of voxels w/ best-fit params within a certain range, but no information about the other constraints imposed)

Thank you for pointing out this omission.

Action: We have included information about the model constraints and seeding in the manuscript at lines 522-526: “Several bounds were imposed on the model fitting: the center X/Y position could not be outside the range of the mapping stimulus ($11^\circ \times 11^\circ$); the σ could not be less than 0.1° or greater than 11° ; gain was bound between 0 and 100; and the exponent was compressive between 0 and 1.”

5. Fig. 5 (and S4) compare something called “FWHM” of pRF coverage maps across conditions – what is this measure the full-width at half-max of? It looks like some ROIs have non-circular/symmetric dashed white lines, which makes it hard to intuit how this quantity is calculated from the data.

This refers to the full-width half-max (FWHM) of the visual field coverage density of each region. When calculated across participants as in the figures, visual field coverage density corresponds to the mean proportion of each subject’s pRFs that overlap with a point in space. The FWHM is drawn as a dashed contour at the half-max value of this density, and is indeed non-circular. We use this as a summary metric of the overall ‘coverage’ of each ROI, as it captures the overall spatial distribution of pRFs while allowing us to maintain and visualize meaningful differences in absolute values of coverage density across regions.

Action: We have further clarified the definition of coverage density and the FWHM calculation in the manuscript (see pg. 11, lines 215-217; caption of **Figure 5**).

6. In Fig. 6/the setup for the behavioral experiment, the authors used a contrast mask based on internal facial features to generate the predicted coverage plots under each stimulus condition. I wonder why the internal feature map was used here rather than the facial silhouette used throughout for pRF modeling? Why not use the same feature map in the pRF modeling itself?

This choice was motivated by a substantial behavioral literature that suggests that face recognition behavior relies on internal face features. We have expanded in the justification of this choice in the manuscript, on page 15, lines 282-295.

We note here that using the silhouette images yields qualitatively similar optimal location predictions for upright and inverted faces, such that both upright and inverted faces are predicted to be better recognized in the left hemifield, and the optimal location for inverted faces is lower than that for upright faces. Still, we believe that use of internal features in this simulation is well aligned with the differing aims of the pRF modeling experiment and the behavioral experiment. Specifically, the former stimulus coding is feature-agnostic, capturing the location of all visual content in the stimuli to provide a task-free estimate of basic spatial processing throughout the visual hierarchy. While this provides a useful approximation of the ‘hardware’ on which task-based spatial integration may operate, prior work (c.f. our 2015 paper) demonstrates that these measurements would be substantially different in face-selective regions had the participants been performing a face recognition task in the scanner. In the simulation for the behavioral experiment, we consider the vast behavioral literature that studies the factors affecting the behavioral face inversion effect (FIE). As noted by Reviewer 2, this literature suggests that the internal features are key for face recognition performance, and affect the FIE. Thus, when simulating the optimal location of faces for behavioral purposes we used internal features in the simulation as a meaningful approximation for the specific recognition task performed by participants.

Action: The revised manuscript now justifies and elaborates on these choices (see pg. 15, lines 282-295).

7. For the behavioral experiment, a few questions about the eyetracking procedures: (a) how is a fixation break defined? (dva threshold?) (b) because the positions were tested blockwise and drift correction was applied online, is it possible subj were just ‘drifting’ to fixate on the center of each face, even during the ITIs?

Because the stimuli in our behavioral experiment were offset by relatively small distances – centered about 1° off-fixation – which are below typical fixation thresholds, trial selection was done manually by examining fixation traces for each trial (360 per participant). This was done once with no subsequent modifications, prior to performance analysis. Trials were excluded any time a participant fixated at >1 position on the screen, or when the overall spread of fixations was increased; both of these heuristics account for potential variability in estimating the absolute position of the eyes. Trials on which only large downward motion was noted were not excluded, as this typically corresponded to blinks. Both the quality assurance (QA) file and the code to generate trial-wise plots of eye position is released at <https://github.com/VPNL/invPRF/tree/main/behavior-analysis>.

Since the drift correction (or, equivalently, inter-trial baseline position measurement) is recorded throughout the experiment, we can examine both the uncorrected data (also generated by the above code) and the distribution of drift correction measurements across

experimental conditions. In both of these outputs, we see no evidence that any individual participants' baseline correction systematically corresponded to the location of the face on the screen. Across subjects, we see no significant effects of the location of the face on the correction values, neither as a main effect ($F(2)=0.60$, $p=0.56$), nor as an interaction with whether the correction was in the X or Y direction ($F(2)=1.12$, $p=0.35$).

Action: We have included additional details about the fixation performance evaluation in the manuscript methods (see pg. 27, lines 635-639).

Baseline drift correction across trial types. The magnitude of correction in the X (light grey) and Y (dark grey) direction is shown for each position condition. Individual participants ($N=9$) are plotted as dots. We see no evident difference in the baseline correction across conditions.

REVIEWER #2

Remarks to the Author:

Review for NCOMMS-20-41124-T 'Holistic face recognition is an emergent phenomenon of spatial integration in face-selective regions'

Poltoratski et al. investigate a timely question with broad relevance: what functional role (if any) does spatial sampling play for face processing in the ventral temporal cortex (VTC)?

Traditionally, the VTC 'what' pathway of visual processing has been characterised as location invariant, but recent years have shown that this view is incorrect. Neural populations in face preferring and other areas have characteristic spatial sampling profiles. However, we don't yet know a lot about the functional consequences of this.

Here, the authors present findings suggesting that the spatial sampling of neural populations in face preferring areas change with the type of stimulus presented. Crucially, these changes in

spatial sampling appear to be at the heart of both, holistic face processing and its breakdown in the face inversion effect (FIE). Specifically, the authors fitted voxel-wise population receptive fields (pRFs) and estimated the visual field coverage of face preferring areas using either upright or inverted faces as mapping stimuli. They found that pRFs and field coverage esp. in fusiform face areas were broader and more foveally centred for upright mapping stimuli. For inverted faces, the fitted pRFs were smaller and shifted downwards, effectively covering less of the upper visual field. Based on this result, the authors predicted that the perceptual FIE should be smaller when stimuli are slightly shifted towards the field coverage for inverted faces (i.e. the lower left). They report results pointing in that direction.

If these results hold, they reveal an important functional role for a novel aspect of spatial sampling in the VTC, an aspect of the visual brain that is virtually ignored by textbook models and still poorly understood. At the same time, they would provide a convincing mechanistic account of holistic integration in face processing and its breakdown in the FIE, a major contribution to a longstanding debate in psychology (c.f. Rossion, 2009).

In my view, the findings by Poltoratski et al. are potentially exciting for a broad range of researchers of the brain and perception. However, I have major concerns regarding methods, analyses and interpretation, which I will list below. I do hope the authors will be able to address these concerns with adequate controls, for which I will provide suggestions.

We thank the reviewer for an enthusiastic and thorough review of this work, and are encouraged by his/her assessment of its significance.

MAJOR:

1. Shifting pRFs versus shifting mismatches between modelled and effective stimulus– or: What’s in a face?

Perceptual studies of the FIE often use stimuli restricted to the *_inner face_*, an oval aperture covering an area from chin to just above the eyebrows, in which the eyes appear at about 2/3 of the overall height (e.g. Van Belle et al., 2010). As acknowledged by the authors, there is plenty of evidence suggesting that responses in face preferring areas are mainly driven by these inner features.

However, the mapping stimuli used here showed a frontal view of more or less the whole head, in which the eyes appear at about 50% of the height. That is, the mapping stimuli extended superiorly to include the full forehead and hair. In the discussion, the authors mention this in passing and point to a supplemental analysis (figure S8) suggesting that the apparent shift in pRFs may be driven by this (but assert that the change in pRF size wouldn’t). If the effective stimulus was limited to the inner face, in the inverted condition modelled stimulus locations will be shifted downwards relative to the effective stimulus. This would necessarily lead to the appearance of a downward shift of pRFs.

This seems a crucial point to me, which deserves much more attention. Doesn't this change the interpretation fundamentally? There may be no shift of pRFs at all, just an artefact of an inadequate model assuming forehead and hair drive responses as well as the inner face. To illustrate the point: If the stimuli would have included the neck instead of hair and forehead, the apparent shift of pRFs may have been the opposite (upwards for inversion). But in that case the apparent shift would not reflect a change in spatial sampling of neural populations, but a change in the offset between the modelled and effective stimulus.

If the effects reported by the authors are truly contributing to the FIE as reported in the literature, they should be observable for the same kind of inner face stimuli (e.g. Van Belle et al., 2010). The ideal way forward would be a control experiment using inner faces as mapping stimuli. Given additional scanning is resource intensive, an alternative suggestion would be to enter the effective stimulus into the model by restricting the apertures for the analysis accordingly. If the downward shift of pRFs vanishes, this aspect of the results has to be interpreted much more cautiously. Crucially, such an analysis would also help to verify the assertion that differences in gain and pRF size cannot be explained by this confound. Do these latter effects persist when entering apertures for the effective stimulus into the model?

The reviewer brings up several important points about how the observed differences in pRF estimates with inversion may depend on the choice of stimulus coding in the model (see also Point 3 from Reviewer 1).

While we agree with the reviewer that stimulus coding impacts estimated pRF parameters and respond to his/her specific questions below, we want to underscore that our results reflect largescale differences in the BOLD responses of face-selective voxels across the visual field and are not driven by our modeling. **Figure 1e** depicts responses in a sample mFus-faces voxel, showing that the magnitude of BOLD responses (in % signal change) to faces at the 25 grid locations used for mapping differs between upright and inverted faces. This pattern of BOLD responses is characteristic of a substantial portion of face-selective voxels; to better illustrate this, we have generated additional voxel plots, now available in the Github repository for this manuscript (<https://github.com/VPNL/invPRF/tree/main/scan-analysis/voxPlots/>).

Further, we have added a new analysis and **Figure S2** illustrating the mean (model-free) BOLD responses to upright and inverted faces across the visual field for each ROI. This data again shows that the average BOLD responses of face-selective areas over the visual field differ with face inversion, and appear smaller in extent and shifted downward when faces are inverted than when they are upright. From these additional analyses, it is evident that the differential pRF model fits for the upright and inverted conditions reflect properties of our raw measurements of cortical responses, and are not contingent on specific modeling choices.

That said, we agree with the reviewer that considering the impact of the chosen stimulus coding is critical for interpreting our results. Indeed, in order to yield a more precise quantification of spatial responsivity than, for example, the mean responses shown in **Figure S2**, the pRF model directly incorporates information about the mapping stimulus. Thus, the specific choices of stimulus coding will affect how parameters are estimated. To further clarify

this relationship, we have implemented a set of new analyses to directly test the effect of stimulus coding on pRF estimates for upright vs. inverted conditions.

New **Figure S12d** shows bootstrapped pRF parameter estimate differences between upright and inverted mapping conditions for several alternate stimulus coding possibilities (**Figure S12c**): (D) Circular disk at the location of the stimuli (as in Kay et al., 2015), (S) face silhouette binary mask, as we report in the main manuscript, (C) Contrast image, which emphasize high-contrast regions in the mapping stimulus, (I) Internal features, which excludes the external features (hair, etc.) of the contrast image as suggested by the reviewer, and (E) Eyes, which excludes all other internal features.

We note that while an internal-features stimulus coding produces more equivalent pRF Y position estimates across conditions in mFus- and pFus-faces as predicted by the Reviewer, this model introduces a Y shift in pRF estimates of V1 and IOG. This result is inconsistent with the measured responses these regions, which do not differ between the upright and inverted conditions (e.g. **Figure 1d**, *scan-analysis/voxPlots/V1/*). Additionally, even though the internal-features model accounts to some degree for the Y shift observed in mFus-faces, it still predicts a smaller pRF size for the inverted vs. upright condition in face-selective areas. Indeed, each of the stimulus coding schemes (D,S,C,I,E) retain differences in pRF estimates between the conditions. Finally, neither the internal features coding scheme (I) nor the other alternative models explain more variance in our data than the silhouette model (**Figure S12e**). We conclude that while stimulus coding is meaningful, the internal-features coding does not yield a parsimonious explanation of the data and the differential responses to upright and inverted faces. We discuss these findings, and how they relate to demonstrated neural selectivity for internal face features, in a new section of the results.

Action: To better illustrate the differences in BOLD responses with face inversion underlying our modeling results, we have updated the Results section (see pg. 6, lines 94-106), added additional example voxel plots to our Github repository (*scan-analysis/voxPlots/*), and added a new summary **Figure S2** showing the mean BOLD responses of each ROI to upright and inverted faces across the visual field. We have also added a new section in the results detailing how the choice of stimulus coding affects the pRF parameter estimates for upright vs inverted faces (pg. 13, lines 243-258) and link to the results in new **Figure S12**.

2. Small apparent increases in pRF size may reflect slightly better fixation compliance in the inverted condition

Observers may have had a harder time complying with the central fixation instruction when upright (rather than inverted) mapping stimuli appeared in the parafovea. This may then lead to the appearance of larger pRFs in that condition. The authors argue that this is unlikely, because they observed no significant differences in fixation task performance. But given the crucial nature of this finding and the potential confound, I think an explicit analysis of fixation compliance would be important.

The methods state that fixation was monitored during the scan with an eyelink 1000. Was this data recorded? If so, the authors could compare the median location and spread of measured gaze locations between conditions. Was gaze position more variable in the upright condition? If so, does the size of this effect match that of the observed increase in pRF sizes? If gaze data was not recorded, the easiest way forward may be a behavioural control experiment in the eyetracking lab, using the same stimuli and task.

The reviewer raises a concern that variable fixation performance between the upright and inverted face conditions may contribute to the observed effects. To address this, we present data from the recorded eye tracking in the scanner (new **Figure S1**).

The reviewer is correct to note that eye position in the scanner was monitored using an Eyelink system. Eye tracking was useful in ensuring participants were alert, awake, and compliant with instructions to fixate. We acknowledge that eye tracking during scan is difficult: calibration failed in 4 of 12 participants and on some fraction of individual runs, and the scanner pulse signal often interfered with the Eyelink recording, resulting in signal loss several times per second that was unrelated to blinks or eye movements. Despite these issues, we present eye tracking data from 8 of 12 subjects, in 77.5% (sd = 14.2%) of their individual runs in the new **Figure S1**.

Following standard preprocessing (blink and aberrant-spike removal, linear and quadratic trend removal), we compared the standard deviation of eye movements in the X and Y direction in the upright, inverted, and blank conditions. We saw no significant main effects of stimulus condition ($F(1)=0.20$, $p=0.67$), nor an interaction with X/Y direction of eye movement ($F(2)=0.76$, $p=0.49$). Numerically, there is more variability in the Y direction compared to the X, perhaps due to blinks, but this is not significant across subjects ($F(2)=2.05$, $p=0.17$). These suggest that fixation performance was not significantly different between upright and inverted conditions.

Action: We have included a new analysis of the scanner eye tracking data (**Figure S1**), confirming that we do not see significant differences in fixation variance across the experimental conditions. These results are referred to on page 6 of the manuscript (line 93).

3. Behavioural experiment: recognition performance may be driven by the eccentricity of inner features

This is essentially the same concern as for the mapping stimuli above. Recognition performance presumably hinges on the inner features. Given the whole-head nature of the stimuli, the eccentricity of these features is reduced in the inverted compared to the upright condition when stimuli are shown at the lower left location. The authors could rule out this confound by repeating the behavioural experiment with inner faces (again, as in Van Belle et al., 2010).

Another suggestion would be to test the FIE for different stimulus sizes. If the breakdown of holistic processing in the FIE is indeed a consequence of smaller pRFs, a strong and counter-intuitive prediction would be that recognition performance for inverted faces is enhanced when they are presented at smaller sizes (at least relative to upright faces). This suggestion may be particularly relevant in case the apparent downward shift of pRFs does not persist when modelling effective stimulus locations (see 1) above).

The reviewer raises two interesting hypotheses based on the behavioral data. First, he/she points out that the internal features of upright and inverted faces in this paradigm differ slightly in their eccentricity, such that inverted face internal features (eyes, nose, mouth) are less eccentric than upright face features at the lower left location, and vice versa in the upper right. While this is the case, it cannot fully account for the observed results. First, we note that not only front-facing, but also $\pm 15^\circ$ rotated viewpoint faces were used in the behavioral experiment, so the location of features was not fully stereotyped throughout a trial. More directly, if eccentricity of the internal features drives performance, we would expect better recognition for upright faces in the upper-right than in the lower-left location. Instead, we see roughly equivalent performance for upright faces at the two positions. Thus, while small eccentricity differences of internal features between the conditions do exist, this is not a parsimonious explanation for the observed pattern of results.

The reviewer also proposes an intriguing hypothesis that recognition performance for inverted faces may be enhanced if they are made smaller, i.e. so that they better ‘fit’ within a reduced field-of-view afforded by smaller pRFs. While it may indeed be the case that smaller faces would be processed more holistically in such a paradigm, we would also expect at the same time a decrease in performance due to retinal acuity and crowding (Martelli 2005, Whitney & Levi 2011). Thus, disentangling these factors of size, retinal acuity, and crowding would pose a considerable, albeit very worthwhile challenge. As we are currently limited in our ability to conduct in-person research due to COVID-19, we present this as a proposed follow-up in the discussion (see pg. 20, lines 427-437).

Action: We added a discussion of why the different eccentricities of the upright and inverted face features in the lower left and upper right cannot simply explain the behavioral results on page 26, lines 606-613. Additionally, we discuss proposed future directions, including a more complete paradigm to evaluate the relationship between face size, acuity, crowding and inversion on recognition performance, on page 20 (lines 427-437) of the discussion.

4. Behavioural experiment: crucial interaction hypothesis not tested

The authors claim that ‘the behavioral FIE (upright minus inverted performance) was significantly reduced in the lower-left position’ (legend of figure 7) and that they ‘showed that the magnitude of FIE can be mitigated by placing faces in an optimal location’ (discussion). However, as far as I see, this interaction hypothesis was not tested. Instead this interpretation seems to rest on the fact that the FIE was significant at the two control positions, but only a non-significant trend was observed at the shifted location. This is a statistical fallacy. Absence

of significance is no significant evidence of absence (esp. when $p = .13$ at a sample size of $n=8$) and a difference in significance provides no significant evidence for a difference (e.g. Gelman & Stern, 2006, The American Statistician). The authors should explicitly test their interaction hypothesis (reduced FIE at lower left compared to other locations). It may be wise to precede this with a power calculation for such an interaction test and increase sample size accordingly.

The reviewer is correct – absence of significance is not significant evidence of absence. As described in the response to Reviewer 1, the lack of ANOVA results in the paper is a mistaken omission, as the test was part of the original analysis code (behavior-analysis/fig7_prfRec2_analysis_plusQA.m). We apologize for this mistake and thank both reviewers for bringing it to our attention. The interaction term is indeed significant (condition x location, $F(2) = 4.57$, $p = 0.027$), as are both main effects.

Action: This statistical result is now reported in the paper (see pg. 17, lines 331-335).

A further concern is regarding the prediction itself. Wouldn't the hypothesis that the lower left is a better match for the inverted field coverage specifically predict increased performance in the inverted condition? Instead, there seems to be a decrease in the upright condition.

Nominally, we do see the best inverted performance in the lower left condition. However, note that our predictions do not take into account visual acuity (which is, of course, much better at the fovea), overtraining of laboratory visual tasks at a central fixation point, nor the increased strain of maintaining fixation while performing the task off-center. Because of this, we focus our comparison on the lower-left versus upper-right retinotopic positions, which should offer comparable visual acuity and difficulty of maintaining covert attention. Nonetheless, it is interesting that despite being off-center the recognition performance for inverted faces in the lower-left location is not substantially different than performance at center, while for upright faces performance in the lower-left is lower than at center.

Action: We add additional clarifying language to this point in the description of the behavioral experiment (see pg. 16, lines 314-318).

MINOR:

5. Noise simulation: betas of the upright condition are scaled to match the observed range of betas for the inverted condition – please clarify whether this was limited to positive values or not and briefly explain the rationale for that

The range is not limited to positive values, although the schematized voxel in **Figure S4** (formerly **S2**) happens to contain all positive betas in the upright condition. The scaling does not change the sign of the data, although additive noise may do so.

6. Noise simulation: the observed range for the inverted condition includes both, signal and noise. However, the final range of the simulated betas - after iterative addition of noise - will be larger than that, correct? If so, please clarify how large the final difference in scale is. A circular procedure with multiple iterations of scaling and adding noise may achieve both, similar scale and noise levels. However, given the results in figure S2c, I don't think this is essential.

The reviewer is correct to note that the scaling step was done prior to iterative noise addition, so it is likely that the range of the simulated data in the original **Figure S2b** was larger after adding noise than what we've prescribed. To address this concern, we updated this simulation to apply the scaling term iteratively, producing a new panel **Figure S4b**. However, as the reviewer (and the results of original **Figure S2c**) predict, the effect of scaling is largely limited to the gain parameter estimate, and this updated simulation yielded similar results to the prior version.

Action: We have performed a new simulation that applies scaling iteratively, and replaced panel C of **Figure S4** (formerly **S2**).

7. Relation to optimal fixation behaviour: Please briefly discuss how your results relate to previous findings showing optimal fixation positions within faces (Peterson & Eckstein, 2012 PNAS), as well as a possible match between spatial and feature sampling in face areas (e.g. Issa & DiCarlo, 2012 JNeuro; de Haas et al., 2016 JNeuro)

Action: We have added additional comments on this literature in the results (see pg. 15, lines 287-290) and discussion (see pg. 397, lines 399-409).

8. Stimulus familiarity: Please briefly discuss the familiarity of participants with the stimulus set and the hypotheses (esp. regarding authors participating in the behavioral experiment)

In total, data from thirteen subjects was collected (one was excluded from analyses because of motion during the scan). Of these, SP, DF, and 6 additional members of the lab (undergraduates, graduate students, and postdocs) had some awareness of the general research line and may have worked with this face set, which has been used by our group for ~15 years. Only author SP was aware of the specific hypotheses of the studies at the time the experiments were conducted; author DF joined only at the analysis stage.

To address the potential confound of author SP's awareness of the study design, we ran the analyses excluding her data (below, left), and replacing her data with that of the subject excluded for scan motion such that N is matched to the original study (below, right). Removing SP's data yields a qualitatively similar pattern of results; replacing this data (thus maintaining the original study's power) yields similar statistical conclusions.

9. Threshold: How was the inclusion criterion $R^2 > 20\%$ chosen?

This criterion was chosen to a) include a sizeable proportion of face-selective voxels in most individual subjects, and b) exclude a sizeable proportion of voxels in primary visual cortex that had inconsistent goodness-of-fit between conditions (bottom left corner of the V1 plot below; this plot follows conventions of original **Figure S1** (now **S3**) but shows the R^2 for every recorded voxel, prior to thresholding).

Action: To provide more detail on our selection of voxels, we have added a new panel to **Figure S3** that illustrates the proportion of voxels in each region that are trimmed (a) by the preset inclusion criteria (minimal/maximal size, maximal eccentricity), and (b) then, by goodness-of-fit R^2 .

10. Line 413: What does '60s' refer to in the context of low-pass filtering? Do you mean high-pass filtering signal with a period shorter than 60s? Sorry, in case I missed something obvious.

We thank the reviewer for catching this typo (we indeed meant high-pass filtering), as well as the errors below.

Action: All errors have been corrected.

11. SI line 11 superfluous 'that' ('match that the range')

12. SI lines 99 and 100 – text seems to swap 'left' and 'right' panels

13. Reporting of decimal points for stats (t-vals etc.) is inconsistent (sometimes two, sometimes three)

14. Figure 5 : please provide an indication of scale (e.g. line showing extent of 1 dva)

15. Line 459 'e.g.' presumably is meant to say 'i.e.'

REVIEWERS' COMMENTS

Reviewer #1 (Remarks to the Author):

I appreciate the authors' careful attention to detail in their response to my comments, and I hope they agree that the manuscript is much improved with this revision. I have no further comments at this time.

Reviewer #2 (Remarks to the Author):

The authors provide a very thorough revision, which addresses all my concerns and I fully recommend publication.

However, I have one remaining suggestion. Most readers will probably not consult the Supplemental Information. I think they should still have a chance to understand that the position of internal features in upright vs inverted faces could account for the observed vertical pRF shift in mFus and pFus (Figure S12d). I appreciate the authors convincing arguments explaining that this can't be the full story. Nevertheless, the finding seems important enough to be featured in the main text.

This could be done with a small edit, e.g. by prepending line 253 along the lines of 'While the shift of internal features between upright and inverted stimuli could account for the observed vertical shift of position preferences in mFus and pFus (Figure S12d), we see that neither [...] is sufficient to explain all observed effects across areas [...]'

Also, the counterargument in the supplemental figure legend would do well to differentiate between V1 and IOG. Surely, an internal feature model wouldn't be expected to apply to V1, while for IOG it very much would.